# Stability of Halocarbons in Whole Air Samples Collected in Stainless Steel Canisters

Tanja J. Schuck[1], Ann-Katrin Blank[1], Elisa Rittmeier[1], Jonathan Williams[2], Carl A. M. Brenninkmeijer[2], Andreas Engel[1], and Andreas Zahn[3]

[1]Goethe University Frankfurt, Germany
[2]Max Planck Institute for Chemistry Mainz, Germany
[3]Karlsruhe Institute of Technology, Germany

**Correspondence:** T. J. Schuck (schuck@iau.uni-frankfurt.de)

**Abstract.** Measurements of halogenated trace gases in ambient air frequently rely on canister sampling followed by offline laboratory analysis. This allows for a large number of compounds to be analysed under stable conditions, maximising measurement precision. However, individual compounds might be affected during sampling and storage of canister samples. In order to assess halocarbon stability in whole air samples from the upper troposphere and lowermost stratosphere, we performed sta-

bility tests using the air sampling unit High REsolution Sampler (HIRES) which is part of the CARIBIC (Civil Aircraft for the Regular Investigation of the Atmosphere Based on an Instrument Container) instrument package. HIRES holds 88 light-weight stainless steel cylinders that are pressurized in flight to 4.5 bar using metal bellows pumps. The HIRES sampling unit was first deployed in 2010, but has up to now not been used for regular halocarbon analysis with exception of chloromethane. The sample collection unit was tested for sampling and storage effects of 28 halogenated compounds. The focus was on compound

stability in the stainless steel canisters during storage of up to five weeks and on the influence of ozone, since flights take place in the upper troposphere and lowermost stratosphere with ozone mixing ratios of up to several hundred ppbV. Most of the investigated (hydro)chlorofluorocarbons and long-lived hydrofluorocarbons were found to be stable over a storage time of up to five weeks and were unaltered by ozone being present during pressurization. Some compounds such as for example dichloromethane, trichloromethane and tetrachloroethene started to decrease in the canisters after a storage time of more than

two weeks or exhibited lowered mixing ratios in samples pressurized with ozone present. Few compounds such as for example tetrachloromethane and tribromomethane were found to be not stable in the HIRES stainless-steel canisters independent of ozone levels. Also growth was observed during storage for some species, namely for HFC-152a, HFC-23, and Halon-1301.

## 1 Introduction

Despite their low atmospheric mixing ratios of only a few hundred parts per trillion or less, halogenated trace gases have a sig-

nificant impact on the Earth's atmosphere. In particular, anthropogenic chlorinated and brominated halocarbons are responsible for stratospheric ozone depletion (Engel and Rigby et al., 2018). Of particular interest are the mixing ratios of such species in the upper troposphere as an entry point for chlorine and bromine into the stratosphere.

The trace gas composition in the upper troposphere and lowermost stratosphere (UTLS) can be analysed from aboard aircraft using in-situ instrumentation or using air sample collection followed by post-flight analysis on the ground. Whereas fast in-flight measurements based on gas chromatography/mass spectrometry provide data of halocarbon and non-methane hydrocarbon mixing ratios in the UTLS at a higher spatial resolution (1–4 min) (Apel et al., 2003; Sala et al., 2014; Bourtsoukidis et al., 2017), the number of species quantified is often limited while the instrumentation is complex. Although canister air sampling typically generates data at a much lower spatial coverage, the post-flight analysis allows a wider range of substances to be measured. Thus, air sample collection is a well-established part of the scientific payload of both campaign-type studies such as for example the HIPPO (Wofsy, 2011) and ATTREX (Navarro et al., 2015; Jensen et al., 2017) projects and long-term projects such as CONTRAIL (Machida et al., 2008) and IAGOS-CARIBIC (Brenninkmeijer et al., 2007; Petzold et al., 2015).

Literature on compound stability in sampling canisters mainly deals with volatile organics. For example, Lerner et al. (2017) found alkanes and alkenes to be stable in electropolished stainless steel canisters in a storage experiment over up to four days, but stability of oxygenated compounds was influenced by the level of humidity. Mixing ratios of aldehydes and monoterpenes were found to decrease significantly in stainless steel cylinders even after short storage times of only a few days but exhibited extended stability in humidified cylinders (Batterman et al., 1998). In a storage experiment covering a period of up to 28 days, Ochiai et al. (2002) found polar VOCs to be more stable at elevated relative humidities while non-polar VOCs were also stable in dry canisters. Better stability was found for VOCs in fused-silica-lined cylinders compared to electro-polished stainless steels cylinders (Ochiai et al., 2002; Hsieh et al., 2003).

The CARIBIC project (Civil Aircraft for the Regular Investigation of the Atmosphere Based on an Instrument Container, www.caribic-atmospheric.com) investigates atmospheric composition from aboard a Lufthansa passenger aircraft equipped with a sophisticated air inlet system (Brenninkmeijer et al., 2007). CARIBIC is a long-term scientific project, employing a comprehensive set of instruments for simultaneous measurements of trace gases and aerosol particles inside a 1.5 t air freight container during regular long-distance flights of the aircraft. Measurement flights take place over 2–4 consecutive days 6–12 times per year. The instrument package consists of remote sensing instruments, fast in-situ instruments and collection of particulate matter and air samples (Brenninkmeijer et al., 2007; Petzold et al., 2015).

CARIBIC air sampling initially was limited to 28 glass flasks per series of flights. Taking advantage of the large spatial coverage of commercial long-distance flights, measurements from these CARIBIC glass flask samples have provided valuable information on the distribution of halogenated trace gases in the upper troposphere and lowermost stratosphere for example for HFC-227ea (Laube et al., 2010), $SF_5CF_3$ (Sturges et al., 2012), perfluorocarbons (Laube et al., 2012), dichloromethane and other short-lived chlorocarbons (Leedham-Elvidge et al., 2015; Oram et al., 2017) and bromocarbons (Wisher et al., 2014), CFC-114 (Laube et al., 2016) and CFC-113a (Adcock et al., 2018). Complementing the two existing sampling units with glass cylinders, the new HIRES unit with 88 stainless steel cylinders was built for the automated collection of whole air samples at a higher frequency in 2010 (Schuck et al., 2012). Since then it has been regularly employed for analysis of greenhouse gases and non-methane hydrocarbons (Baker et al., 2016; Li et al., 2018), but it has not been used for dedicated measurements of halogenated trace gases.

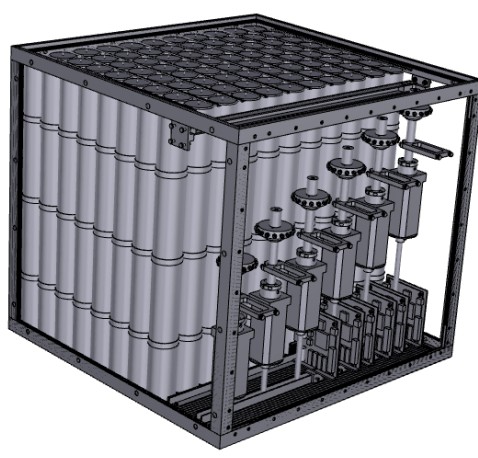

**Figure 1.** Schematic view of the 88 sample cylinders inside HIRES and the six multi-position valves. For simplicity all tubing is omitted to illustrate only the positioning of these main components. Drawing: Laurin Merkel.

Measurements of halocarbons in the tropopause region are sparse. CARIBIC air samples are collected on a regular basis and therefore complement measurements from research aircraft campaigns such as for example the ATTREX (Navarro et al., 2015) or TACTS/WISE/PGS projects (Keber et al., 2019). In order to explore the potential of the HIRES unit for halocarbon measurements, we intensively tested the sampling unit in a series of laboratory experiments. In particular, we investigated the stability of 28 compounds during storage between sampling and analysis over typical processing times. In addition, we investigated the influence of ozone, which may affect sampling of reactive species in the lowermost stratosphere.

## 2 Technical Details

### 2.1 The HIgh REsolution Sampler

HIRES holds 88 light-weight cylinders made of stainless steel (wall thickness 0.25 mm), each with a volume of 1 L, total weight is 43 kg. Samples are pressurized to 4.5 bar in flight at pre-set time intervals using two metal bellows pumps (Senior Aerospace Metal Bellows, 28823-7). Coarse particles are filterd by a 2 $\mu$ filter (Swagelok SS-4F-2). Each cylinder is connected to one of six 16-position valves (Valco, EMT4LSD16MWE) by which the samples are selected. In a laboratory setting, up to six samples can be pressurized simultaneously with identical air mixtures. Figure 1 shows a schematic view of the cylinders and valves.

Before a flight, HIRES undergoes leak testing with ambient air passed through a molecular sieve, but cylinders are not preconditioned. On take-off, cylinders will usually hold remnant air from the last research flight or from the leak test. The reason is that due to mechanical stability of the thin-walled flasks they should not be evacuated to absolute pressures below

600 mbar. Tests during the construction phase and monitoring based on NMHC measurements during the first years of operation of the sampler have shown that eight iterations of flushing do reliably dilute remnants of previous fillings of tropospheric air. In flight, canisters are therefore flushed with ambient air eight times, this is achieved by filling a flask to 4 bar followed by venting for 20 s. After this time ambient pressure is reached which aboard the aircraft at flight altitude is $\sim 700$ mbar.

After that, canisters are eventually pressurized to 4.5 bar. The total time needed for this procedure is $\sim 4$ min of which the final pressurization takes 10–20 s. During each flushing about 20 % of the air from the previous filling remains in the sample canister. Depending on ambient pressure, 70–80 % of the air is from the final pressurization stage. The sampling period is defined as the time interval during which at least 97 % of the sample air was collected. This comprises the last three of the eight flushing iterations and the final pressurization stage, adding up to a total sampling time of 1–2 min. Data from continuously measuring

instruments, such as ozone ($O_3$) (Zahn et al., 2012) and carbon monoxide (CO) (Scharffe et al., 2012), are integrated over the sampling intervals for comparison with canister measurements.

    Upon return of the instrument container, the sample collector is de-installed, and post-flight gas chromatography analyses are performed in the laboratory for greenhouse gases (Schuck et al., 2009) and non-methane hydrocarbons (Baker et al., 2010). If a halocarbon analysis is performed it is usually last in a series of measurements and takes place approximately 3 to 5 weeks

after the flight. The duration of the long-term storage test time of 8 weeks was deliberately chosen beyond this period.

## 2.2   Halocarbon Measurements

The halocarbon measurements are based on adsorption-desorption gas chromatography / mass spectrometry (GC/MS) (Hoker et al., 2015; Schuck et al., 2018). As halocarbon mixing ratios in the atmosphere are in the ppt range, pre-concentration of the sample air prior to gas chromatographic separation and detection is required (Obersteiner et al., 2016). In addition, the

sample air is dried by passing over a heated (80 °C) tube filled with magnesium perchlorate $Mg(ClO_4)_2$. Mixing ratios are therefore reported as dry mole fractions. All tubing is heated to avoid condensation of moisture (relevant for HIRES only for tropospheric samples) and to minimize wall losses. Following the drying unit, the sample flow is directed through a 1/16" stainless steel sample loop (inner diameter 1 mm) filled with HayeSep D over a length of 10 cm. During the adsorption phase the sample loop is kept at -80 °C (Stirling Cooler, Global Cooling, M150). Depending on the pressure of the HIRES air samples

at the time of measurement, the enrichment flow is set to either 100 ml/min or 150 ml/min controlled by a mass flow controller mounted behind the sample loop in order to exclude sample contamination from the controller. The enriched sample volume is determined by monitoring the pressure inside a 2 x 2 L reference volume which gets evacuated prior to sample enrichment. Typically a volume of 0.8–1.0 L of air is used. For desorption, the sample loop is heated to approx. 200 °C for 4 min while the carrier gas flow is directed through it (Helium grade 6.0 (Praxair), Purification System: Vici Valco HP2).

Gas chromatography is performed with an Agilent 7890A instrument equipped with a 7.5 m pre-column and a 22.5 m main column (both GasPro PLOT, inner diameter 0.32 mm). The temperature program of the GC starts at 50 °C kept for 2 min after which the oven is heated to 95 °C at a rate of 15 °C/min. Then it is heated to 135 °C at 10 °C/min, and finally to 200 °C at a rate of 22 °C/min. This temperature is kept for another 2.95 min. The complete runtime adds up to 17.95 min. Backward flushing of the pre-column is started after 12.6 min to avoid contamination of the subsequent chromatographic run with high-boiling

substances. Mass spectrometric detection is with a quadrupole (QP) mass spectrometer (Agilent 5975C) operated in selected ion monitoring mode, scanning pre-selected masses at a given retention time. Ionisation is via electron impact at 70 eV. Data from a time-of-flight-mass spectrometer operated in parellel to the QP-MS are not discussed here. The experimental setup was described in more detail by Hoker et al. (2015) and Obersteiner et al. (2016).

5    HIRES samples are measured relative to a laboratory standard which has been collected cryogenically at Jungfraujoch (Switzerland) in December 2007. It is compared to a tertiary standard of the Advanced Global Atmospheric Gases Experiment (AGAGE) network monthly and has been re-calibrated versus several AGAGE standards in December 2018. Drift of the working standard can thus be excluded. Mixing ratios are reported in ppt on Scripps Institution of Oceanography (SIO) scales except for Trichloroethene, Dibromochloromethane, Tetrachloroethene, and Tribromomethane (reported on scales defined by
10   University of Bristol, University of East Anglia and the National Oceanic and Atmospheric Administration). Mixing ratios of CO and Ozone from CARIBIC in-flight measurements are reported in ppb per Volume (ppbV). Details of the respective calibration of both instruments were published by Scharffe et al. (2012) and by Zahn et al. (2012).

## 2.3   Storage Experiments

To test a possible influence of ozone on reactive halocarbon species, HIRES sample canisters were pressurized in the labo-
15   ratory with a mixture of a well-characterized laboratory standard and synthetic air. The standard was filled with an oil-free compressor at Taunus Observatory (50.22 ° N, 8.44° E , 825 m.a.s.l) in 2015 and contained typical tropospheric mixing ratios of halocarbons. During pressurization it had been dried using magnesium perchlorate. This standard was chosen for the storage experiments, as air from the UTLS usually contains water vapour mixing ratios of less than 100 ppmV.

Figure 2 shows the set-up used for the pressurization of HIRES for storage tests. Samples could be filled with either the standard gas, synthetic air or a mixture of both. Contrary to the set-up in flight, when the HIRES cylinders are filled with ambient air pressurized by the metal bellows pumps, in this set-up they are pressurized directly from high pressure gas cylinders.
On path I the synthetic air flow passed a quartz glass tube exposed to UV light from a mercury lamp generating ozone. On average, $\sim 1200$ ppbV of ozone was generated as determined by a separate measurement (API Photometric Ozone Analyzer T400), but it was not monitored during the filling of the cylinders. After mixing of the ozone enriched synthetic air with the standard gas (path III), ozone mixing ratios were approx. 400–600 pbbV, being characteristic of the lowermost stratosphere as typically sampled by the CARIBIC aircraft. This procedure was chosen, because generating ozone directly in the flow of the
standard gas could have altered mixing ratios of the reactive halogenated compounds contained in the standard.

For a short-term storage test, six individual canisters were pressurized consecutively to an absolute pressure of 4 bar using 1.7 bar of synthetic air and 2.3 bar of the standard gas. For three of the canisters, the synthetic air was enriched in ozone by directing its flow via the UV lamp. All samples were analysed one day after filling and again one week later.

Because of the large number of cylinders, measurement of all samples of one CARIBIC flight series takes several days,
depending on the type of analysis performed. The halocarbon measurements described here add up to approx. 53 hours total measurement time, not including blank measurements and preparation work. In addition, HIRES needs to circulate between

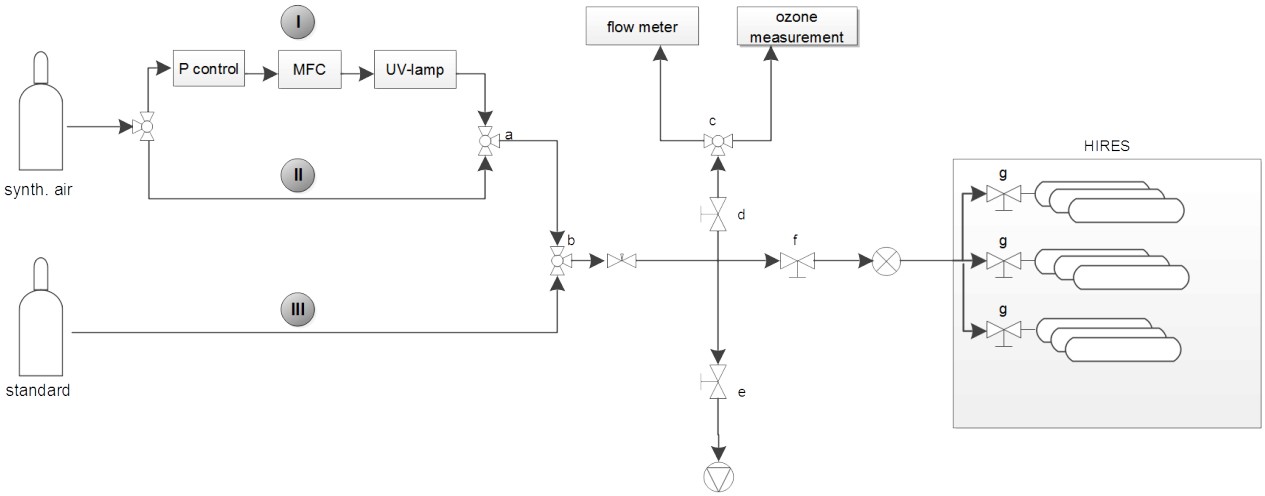

**Figure 2.** Schematic view of the gas flows for filling HIRES canisters for the storage experiments. Samples could be pressurized with either pure air from a standard gas bottle (flow path III), pure synthetic air (path II), or with a mixture of both. Alternatively, the flow of synthetic air could be directed to pass an ozone generating UV lamp (path I).

different laboratories in different institutions, therefore the time between in-flight sampling and post-flight sample analyses can be much longer than one week.

Canisters have a volume of 1 l and were pressurized up to 4 bar. Due to mechanical stability of the thin-walled flasks they
may not be evacuated during the measurement. For a corresponding long-term storage test with six measurements of one gas mixture the sample volume of one cylinder would not be sufficient. Therefore, six cylinders were simultaneously filled, and on each measurement day, the next canister of such a series was measured. Canisters were pressurized to an absolute pressure of 4 bar with one of the following gas mixtures:

- synthetic air (4 bar)

- synthetic air (1 bar) + standard (3 bar)

- synthetic air (1.7 bar) + standard (2.3 bar)

- synthetic air (1.7 bar) + ozone + standard (2.3 bar)

- synthetic air (4 bar) + ozone

With each of the mixtures two subsets of six canisters each were pressurized simultaneously, thus containing identical com-
position. One canister out of each subset was analysed after a storage time of 1, 8, 15, 29, 51, and after 57 days. The full measurements series for the long-term storage test comprises analyses of 60 individual canisters. Between measurements, the

sampling unit was stored in an air conditioned laboratory at temperatures around 22 °C. The air conditioning uses water and is therefore not expected to adversely influence halocarbon analysis.

Assuming the synthetic air to be free of any of the compounds of interest, mixing ratios in the canisters should be identical to the original mixing ratios weighted by the relative contribution of each gas, i. e. trace gas mixing ratios in the HIRES canisters should be about 75 % of the original mixing ratios of the standard in the case of 1 bar of synthetic air mixed with 3 bar of the standard gas and about 57.5 % in the case of 1.7 bar of synthetic air mixed with 2.3 bar of the standard gas. Measurements of the pure synthetic air, however, revealed contamination of the synthetic air with carbonyl sulfide (12.8 ppt), chloromethane (24.3 ppt), HFC-152a (2.8 ppt), and tetrachloroethene (1.1 ppt). This needs to be taken into account for calculation of the trace gas mixing ratios expected in the HIRES canisters after mixing synthetic air and standard gas. To reduce uncertainties related to the mixing of synthetic air and standard gas, which are mainly caused by the uncertainty of the pressure readings of approx. 0.1 bar, all compounds were evaluated relative to CFC-12 ($CCl_3F$). CFC-12 was found to be stable in the canisters.

## 3 Results

### 3.1 Ozone interference and short-term stability

The short-term storage test was performed to investigate the influence of ozone being present during pressurization. HIRES is usually operated in the UTLS region and ozone mixing ratios are commonly up to 800 ppbV, depending on flight route and season. Figure 3 shows as examples the results of the test for HFC-134a ($CH_2FCF_3$) (panel (a)) and dichloromethane ($CH_2Cl_2$) (panel (b)). Plotted is the ratio of the mixing ratio of the respective substance to CFC-12 ($CCl_3F$) as described in the previous section. The solid black line represents the value expected from the known mixing ratios of the standard gas. For each measurement day a daily precision value was calculated from the variability of the measurements of the standard. This expected daily uncertainty range is represented by the grey error bars. Solid coloured lines stand for canisters pressurized with synthetic air not treated by the UV-lamp, dashed coloured lines represent canisters pressurized with the synthetic air passing the UV-lamp and thus ozone being present. Error bars for the individual data points are not shown as they overlap and merge into one undistinguishable error bar. However, if the symbols fall within the uncertainty range indicated by the grey error bars of the expected value this means, that they agree within 2-$\sigma$ with the expected value.

For HFC-134a, the ratio of mixing ratio agrees with the expected value within the experimental uncertainty on the first day and on day 8. For dichloromethane, samples influenced by ozone (dashed lines) show a significantly lower ratio to CFC-12 than expected. No systematic change from day 1 to day 8 was measured for either of the two compounds. While dichloromethane was stable over a storage time of one week, it was influenced by ozone and exhibited depleted mixing ratios in the canisters already one day after pressurization. It can thus not be reliably measured from HIRES canisters from the lowermost stratosphere.

Substances that were found to be depleted in HIRES canisters when pressurized in the presence of ozone were: dichloromethane ($CH_2Cl_2$), trichloroethene ($C_2HCl_3$), tetrachloroethene ($C_2Cl_4$), and dibromochloromethane ($CHBr_2Cl$). Carbonyl sulfide (COS) showed higher mixing ratios in samples that had been exposed to ozone. While there is clear evidence, that the mixing ratios of these substances are modified in the canisters when they are pressurized at elevated levels of ozone, it is not

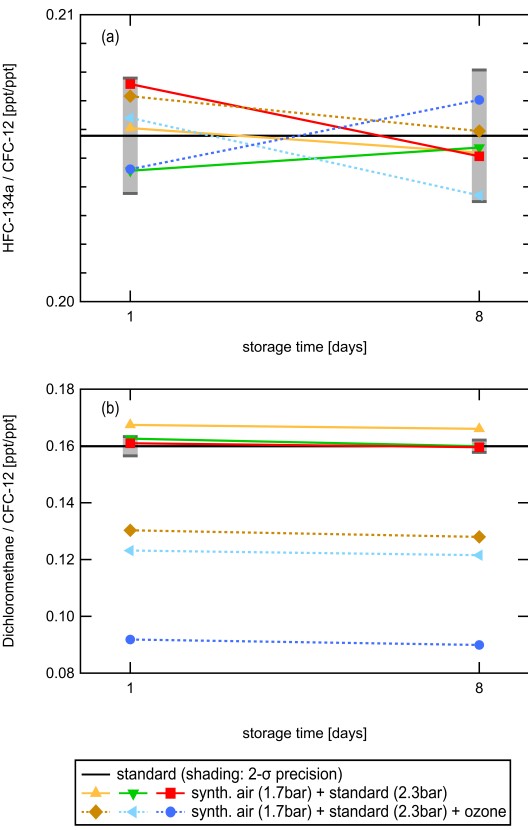

**Figure 3.** Results of the ozone and short-term storage test for HFC-134a (a) and dichloromethane (b). Shown is the ratio of mixing ratios of the respective compound relative to that of CFC-12 to cancel dilution uncertainties. The solid black line represents the value expected from direct measurements of the standard gas and the synthetic air. Shading indicates the 2-$\sigma$ experimental uncertainty range around the expected value, error bars of individual data points are omitted for clarity.

possible from these experiments to deduce an ozone threshold above which results become unreliable. We will thus consider all UTLS samples characterized as stratospheric by ozone mixing ratios above the respective ozone chemical tropopause, potential vorticity or low mixing ratios of nitrous oxide as not suited for post-flight analysis of these compounds in samples from the current HIRES sampling unit.

5      It should, however, be noted that the experiment does not adequately mimic stratospheric conditions. In the laboratory tests presented here, the reference gas is mixed with the ozone enriched synthetic air during the filling procedure. In flight, stratospheric air masses with high ozone levels will be at some state of mixing and in a continuous chemically processing. In addition contact with hot surfaces such as inside the metal bellows pumps will destroy ozone.

Among the substances influenced by ozone, dibromochloromethane (CHBr$_2$Cl) additionally showed decreasing mixing ratios already after one week of storage (depleted by 94 %), while bromomethane (CH$_3$Br, +46 %) and chloromethane(CH$_3$Cl,

+14 %) were found to grow. Trichloroethene ($C_2HCl_3$) exhibited a variability which did not allow to draw stringent conclusions in the short-term storage test. Tribromomethane ($CHBr_3$), which was unaffected by ozone, was depleted by 70 % after one week of storage. Table 1 summarizes these results.

## 3.2 Long-term stability

The long-term storage test comprised measurements of pressurized canisters after storage times of 1, 8, 15, 29, 51, and 57 days. While for the short-term test, individual canisters were measured on day 1 and day 8, this was not possible for the long-term test. For the long-term test, six cylinders were simultaneously filled, and on each measurement day, the next canister of such a series was measured. Thus, it cannot be fully excluded that stability might not only depend on the substance investigated, but it might be a feature of an individual canister, for example related to the quality of welding seams.

Figure 4 shows as an example results of the long-term storage test for HFC-134a, dichloromethane and HFC-152a. As before, shown is the ratio of mixing ratios of the respective substance and CFC-12. The black line indicates the expected ratio and the grey shaded area represents the experimental 2-$\sigma$ uncertainty range. Solid lines are for measurements of the gas mixture without ozone, dashed line for those with ozone. Error bars for individual data points are not shown. Like most long-lived halogenated tracers, HFC-134a variability is smaller than the measurement precision and measured mixing ratios agree within $2\sigma$ with the expected value.

Some substances that were found to be stable during the one week short-term test decreased after longer storage times, for example dichloromethane shown in Figure 4(b). A similar behaviour was observed for trichloromethane ($CHCl_3$), tetrachloromethane ($CCl_4$), trichloroethene ($C_2HCl_3$), tetrachloroethene ($C_2Cl_4$), tribromomethane ($CHBr_3$), and bromochloromethane ($CH_2BrCl$). Measurements of these compounds should not be evaluated for the HIRES canisters if analysis takes place later than two weeks after sample collection as the long-term test indicates changes of mixing ratios start to occur after that period. In general, the decrease during long-term storage seems to be independent of the influence of ozone, although the gas mixture that shows the largest depletion (light blue dashed line, mostly cut off in Figure 4) did contain ozone. Ozone could not be monitored during the pressurization of the samples. It can therefore not be excluded that this gas mixture may have been exposed to a different amount of ozone than the one represented by the dashed brown line which could have caused the stronger depletion.

Panel (c) of Figure 4 shows the result of the long-term storage test for HFC-152a ($CH_3CHF_2$). While mixing ratios measured on day 1 and 8 were within the expected range, they started to significantly increase after storage day 15. This also occurred for samples influenced by ozone and independent of the gas mixture. A similar result was found for Halon-1301 ($CBrF_3$) and HFC-23 ($CHF_3$). HFC-23 is known to degas from certain materials, thus degassing from valve seals might be a possible source. The rotors of the Valco multi-position valves are made of Valcon E, a polyaryletherketone/PTFE composite, therefore degassing of fluorinated compounds could occur. Of the shorter-lived compounds investigated bromomethane ($CH_3Br$) exhibited an increase in the long-term storage experiment. Table 1 summarizes the results of the long-term and the short-term test.

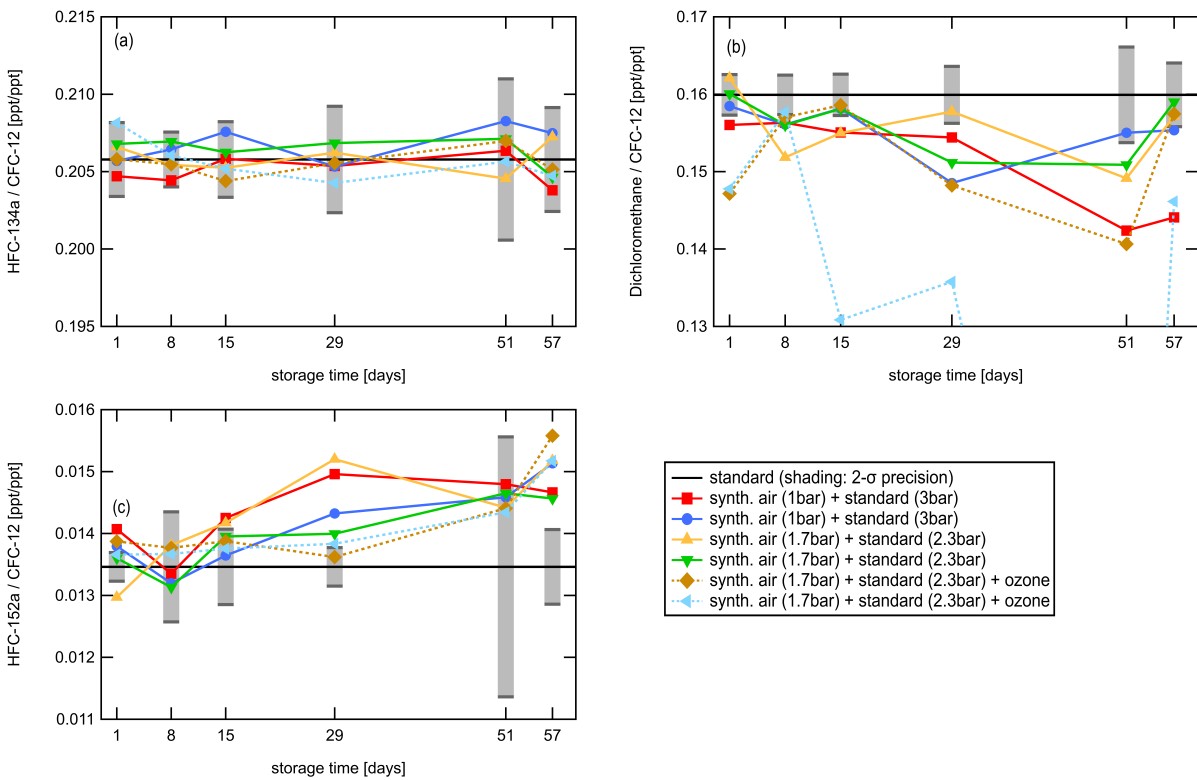

**Figure 4.** Results of the ozone and long-term storage test for HFC-134a (a), dichloromethane (b), and HFC-152a (c). Shown is the ratio of mixing ratios of the respective compound relative to that of CFC-12 to cancel dilution uncertainties. For dichloromethane the vertical scale has been adjusted cutting off samples that were strongly depleted. The solid black line represents the value expected from direct measurements of the standard gas and the synthetic air. Shading indicates the 2-$\sigma$ experimental uncertainty range around the expected value, error bars of individual data points are omitted for clarity.

### 3.3 Air samples from the UTLS

As an example of air collected in the atmosphere under real conditions, samples from CARIBIC flight 544 which took place on 22 March 2018 travelling from Munich (Germany) to Denver (US) were analysed. These measurements were performed approximately 5 weeks after the flight, thus only substances that were shown to be stable in the long-term stability experiment are expected to yield reliable results. Figure 5 shows a time series of ozone and CFC-12 , which was found to be stable in all experiments, and of dichloromethane, which was found to be depleted in samples pressurized at elevated ozone and decreased with time during storage.

CFC-12 anticorrelates with mixing ratios of ozone and this is also found for the other long-lived compounds which were stable in the storage experiments. Such a behaviour is expected, because ozone-rich stratospheric air masses are aged and

**Table 1.** Results of ozone interference, short-term (8 days) and long-term (57 days) storage tests. Arrows indicate whether mixing ratios were increasing ( ↗ ) or decreasing ( ↘ ). Numbers indicate the maximum deviation measured for substances not stable in HIRES cylinders.

| substance | | stable short-term | stable long-term | influenced by O$_3$ |
|---|---|---|---|---|
| CFC-12 | CCl$_3$F | X | X | |
| CFC-11 | CCl$_2$F$_2$ | X | X | |
| HCFC-22 | CHClF$_2$ | X | X | |
| HCFC-141b | CH$_3$CCl$_2$F | X | X | |
| HCFC-142b | CH$_3$CClF$_2$ | X | X | |
| HFC-125 | CHF$_2$CF$_3$ | X | X | |
| HFC-134a | CH$_2$FCF$_3$ | X | X | |
| HFC-143a | CH$_3$CF$_3$ | X | X | |
| HFC-152a | CH$_3$CHF$_2$ | X | ↗ | |
| HFC-23 | CHF$_3$ | X | ↗ | |
| HFC-227ea | CF$_3$CHFCF$_3$ | X | X | |
| HFC-245fa | CHF$_2$CH$_2$CF$_3$ | X | X | |
| HFC-32 | CH$_2$F$_2$ | X | X | |
| Halon-1211 | CBrClF$_2$ | X | X | |
| Halon-1301 | CBrF$_3$ | X | ↗ | |
| Halon-2402 | C$_2$Br$_2$F$_4$ | X | X | |
| Chloromethane | CH$_3$Cl | ↗ | ↗ | |
| Dichloromethane | CH$_2$Cl$_2$ | X | ↘ | ↘ |
| Trichloromethane | CHCl$_3$ | X | ↘ | ↘ |
| Tetrachloromethane | CCl$_4$ | ↘ | ↘ | |
| 1,1,1-Trichloroethane | CCl$_3$CH$_3$ | X | X | |
| Trichloroethene | C$_2$HCl$_3$ | ↘ | ↘ | ↘ |
| Tetrachloroethene | C$_2$Cl$_4$ | X | ↘ | ↘ |
| Bromomethane | CH$_3$Br | ↗ | ↗ | |
| Tribromomethane | CHBr$_3$ | ↘ | ↘ | |
| Bromochloromethane | CH$_2$BrCl | X | ↘ | |
| Dibromochloromethane | CHBr$_2$Cl | ↘ | ↘ | ↘ |
| Carbonyl sulfide | OCS | X | ↗ | ↗ |
| Sulfuryl fluoride | SO$_2$F$_2$ | X | X | |

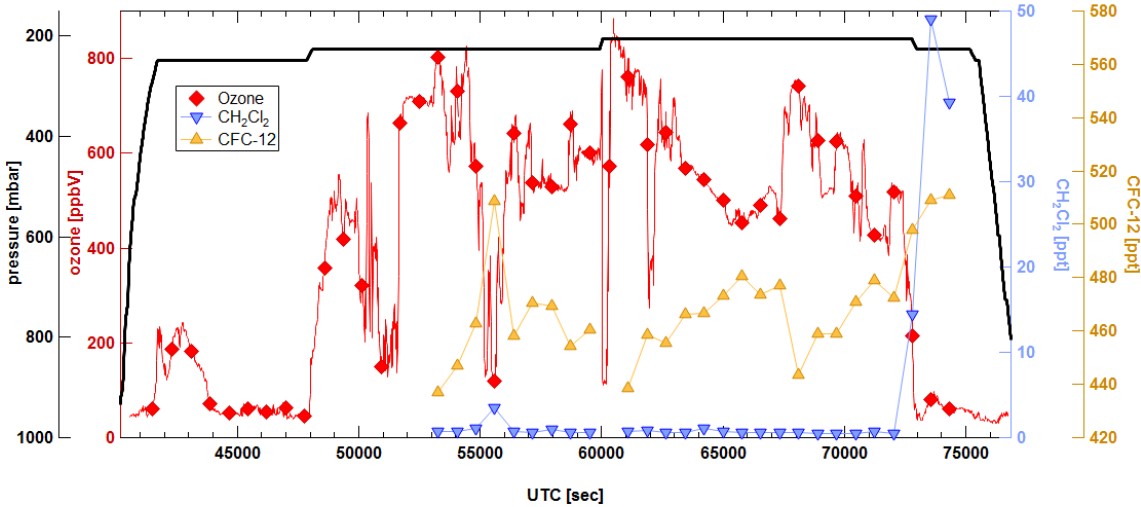

**Figure 5.** Time series of ozone (red), CFC-12 (yellow) and dichloromethane (blue) during a flight from Munich to Denver on 22 March 2018. Ozone high resolution data represented by the red line were integrated over the sampling period of each whole air sample (red diamonds).

should contain lower mixing ratios depending on a substance's stratospheric lifetime and transport pathway. Three of the canisters analysed from this flight were collected in tropospheric air masses characterized by lower mixing ratios of ozone levels. Mixing ratios of CFC-12 measured in these samples are around 510 ppt, consistent with current tropospheric mixing ratios observed at ground sites (Schuck et al., 2018). Similarly consistent numbers are measured for the other compounds expected to be stable in the canisters according to the storage experiments.

Dichloromethane mixing ratios are below 1 ppt in most samples, close to the limit of detection of 0.4 ppt as derived from the 3-fold noise level. Only three samples have mixing ratios above 5 ppt, they are at the same time characterised by higher mixing ratios of CFC-12 and low mixing ratios of ozone which is indicative of tropospheric air. Leedham-Elvidge et al. (2015) reported stratospheric dichloromethane mixing ratios measured from glass samples collected during CARIBIC flights in 2001/2002, 2009/2010 and 2011/2012 to vary around 5–35 ppt. The mixing ratios measured from the HIRES stainless steel cylinders from CARIBIC flight 544 are significantly lower, which is consistent with the storage test results. A similar behaviour was found for all compounds which were not stable in the storage tests.

In the tropospheric air samples, dichloromethane varied between 14 ppt and 49 ppt. This agrees with mixing ratios in tropospheric samples in the dataset presented by Leedham-Elvidge et al. (2015) which were up to 65 ppt with an increase observed from 2006 through 2012, but is somewhat lower than mixing ratios at the ground in March 2018 (Schuck et al., 2018) which would be consistent with the result from the storage test, that dichloromethane is not stable in HIRES cylinders long-term.

Figures 6 and 7 show correlations of CFC-12, HFC-134a and dichloromethane with ozone and CO, respectively. For the stable long-lived compounds CFC-12 and HFC-134a a tight correlation with ozone ($r^2 = 0.98$ and $r^2 = 0.99$) and with CO

($r^2 = 0.80$ and $r^2 = 0.79$) is found. For one tropospheric sample there is no corresponding integrated CO mixing ratio, because it was collected during one of the regular in-flight calibration phases of the CO instrument.

Dichloromethane does not correlate with ozone nor with CO in the stratosphere (panel (c) in Fig. 6 and 7), but it does show a correlation with CO for samples with CO mixing ratios of more than 40 ppbV with a value of $r^2$ of 0.99. Although the value of $r^2$ has a limited meaning owing to the small number of samples, this might be an indication that despite the analysis taking place within five weeks after the flight and dichloromethane being found to decrease with storage time in HIRES canisters, results may still reflect initial mixing ratios. If the decrease rate was the same for all tropospheric samples,
the correlation of dichloromethane with CO might still persist, even if mixing ratios of dichloromethane have decreased. A similar behaviour was found for tetrachloroethene and trichloromethane and for the tropospheric samples for trichloroethene and for dibromochloromethane. The latter compound was below its detection limit in all stratospheric samples, trichloroethene in several of them.

## 4   Conclusions

In order to assess the potential of halocarbon analysis from samples collected with the HIRES unit from the CARIBIC instrument package, the sample collection unit was intensively tested focusing on compound stability in the stainless steel canisters and the influence of ozone. Sampling during CARIBIC flights takes place in the upper troposphere and the lowermost stratosphere with ozone mixing ratios of up to several hundred ppbV. Therefore samples were pressurized with a mixture of a dry standard gas, containing typical tropospheric mixing ratios of a wide range of halogenated hydrocarbons, and synthetic air. The
synthetic air could be enriched in ozone by passing an ozone-generating UV lamp. Final ozone mixing ratios were estimated to range from 400 ppbV to 600 ppbV. This is representative of the mixing ratios typically encountered at flight levels in the lowermost stratosphere. Several short-lived halocarbons were found to be depleted in canisters pressurized in the presence of ozone. COS was found to exhibit higher mixing ratios in this case.

    In one experiment samples were analysed one day after pressurization and again after a storage time of one week. While
bromomethane and chloromethane were found to grow already after this short period, tribromomethane and trichloroethene had decreased, tetrachloromethane was found to be stable but its mixing ratio was significantly below the value expected. Of the 28 compounds investigated, 23 were found to be stable over storage of up to one week.

    This changed in the long-term stability test which was conducted over up to 57 days. All compounds influenced by high levels of ozone were found to show the same behaviour (decreasing or increasing) during the long-term test. In addition,
dichloromethane, trichloromethane, tetrachloroethene, tetrachloromethane, and bromochloromethane also showed a tendency to decrease after storage for longer than two weeks. HFC-152a, HFC-23, and Halon-1301 started to increase significantly. The tests showed, that for a number of halogenated tracers reliable measurement results can only be achieved if measurements are performed within few days after a flight. If this is not possible, which for CARIBIC samples is often the case as they circulate several laboratories at different institutions, results must be interpreted with care.

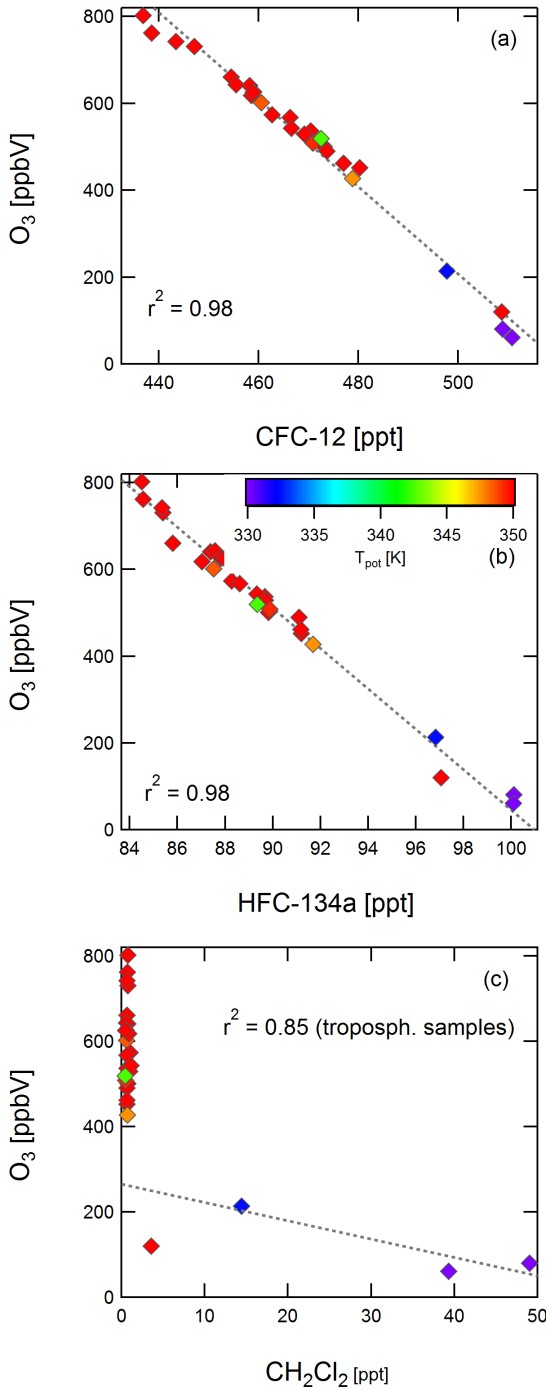

**Figure 6.** Correlation of CFC-12 (a), HFC-134a (b), and dichloromethane (c) with ozone. Colour coding is by potential temperature. Values of $r^2$ are given for the correlation of all samples for CFC-12 and HFC-134a, and for three tropopsheric samples in the case of dichloromethane.

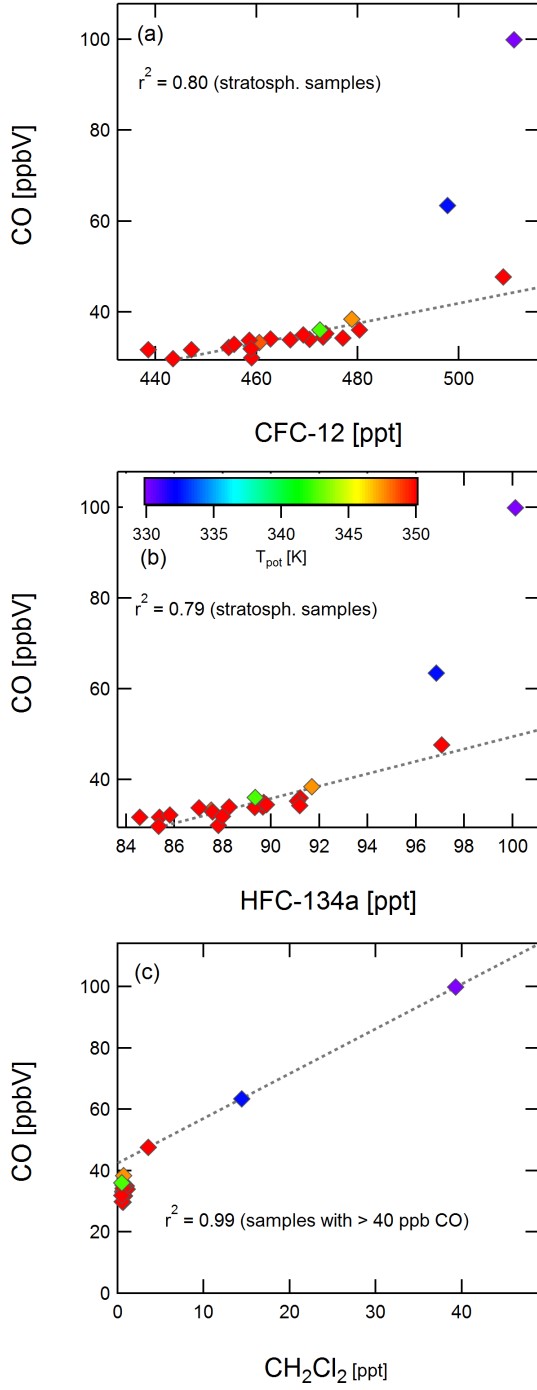

**Figure 7.** Correlation of CFC-12 (a), HFC-134a (b), and dichloromethane (c) with CO. Colour coding is by potential temperature. Values of $r^2$ are given for the correlation of stratospheric samples for CFC-12 and HFC-134a, and for three samples with a CO mixing ratio above 40 ppb in the case of dichloromethane.

Measurements of samples from a CARIBIC flight in March 2018 that took place about five weeks after the flight confirmed the results from the stability tests. Mixing ratios of compounds found to decrease in the stability tests were in general very low,

5 often below their respective detection limits. Also the results of the ozone test were confirmed, as mixing ratios of compound found to be sensitive to ozone were low in canisters sampled in the stratosphere at high ozone mixing ratios. With the current HIRES sampler it is not possible to study for example vertical gradients of mixing ratios above the tropopause for short-lived species. Compounds that had grown during the storage test were not evaluated.

Currently, we are in the process of constructing a second high resolution air sampler for use inside the CARIBIC container.

10 Based on the measurements presented here, close attention will be given to the manufacturing of the stainless steel cylinders which will be made of electro-polished stainless steel foil and welding will be done under vacuum.

*Data availability.* Data are available from the corresponding author upon individual request.

*Competing interests.* The authors declare that there are no competing interests.

*Acknowledgements.* The authors acknowledge the contribution of technical staff performing regular maintenance of the CARIBIC container

and handling of the air sampling unit. We also would like to thank Martin Vollmer (EMPA) for calibration of the laboratory standard and Laurin Merkel for drawing the HIRES sampling unit.

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
