# Peer review of "Stability of Halocarbons in Whole Air Samples Collected in Stainless Steel Canisters"

_Atmospheric Measurement Techniques, 2019_

## Referee Comment (RC1) · Anonymous Referee #1 · 19 Jul 2019

**Review of Shuck et al.; Stability of halocarbons in whole air samples from the upper troposphere and lowermost stratosphere**

Shuck et al., present the results of short and long-term storage tests conducted on stainless steel whole air samples that together form the HIRES sampling system. HIRES is part of the CARIBIC instrument package. The paper focuses on the stability of a range of halocarbons, many of which are important both from a climate and stratospheric ozone perspective. Given the significance of a number of these compounds, and the importance of whole air sampling as a means of complimenting the existing measurement framework, this work fits within the scope of AMT. The paper is, in general, well written and should be published after some revision.

Main comments:

Figure 3. There seems to be some confusion regarding which species are presented. Panel B shows results for dichloromethane, whereas the text talks about tetrachloroethene (P6 L30, P7 L3). I'm assuming this is simply a mix-up, but it makes the entire section rather difficult to understand. With regards to Figure 3, it is also not clear to me as to why the dichloromethane/CFC-12 ratio of the 'orange' sample is higher than the other two samples filled with the exact same ratio of standard/synthetic gas. I suspect that all three are within the measurement uncertainty. In which case it might be good to find some way of showing the error bars which are currently omitted.

While there were several very interesting results presented, I did not feel that the authors provided much in the way of reasoning for a number of their findings. For instance, why did bromomethane and chloromethane increase in the cylinders over the length of the short-term (and long-term) storage tests? Presumably due to production as a result of the decomposition of more complex chlorinated compounds? The same can be said of H-1301, which would appear to be a strange result. I would not expect to see complex chemical mechanisms in this sort of paper, but it would be useful to have some brief discussion of these points.

Technical corrections:

I think the title needs to be revised. At current, it suggests that the storage tests were conducted on samples collected in the UTLS, while in fact they were conducted using samples prepared in the laboratory.

P1 L1. 'Halogenated halocarbons of' should be changed to 'halocarbons in'.
P1 L17. The sentence needs extending. Suggest 'growth was observed during storage for some compounds,…'
P1 L21. Delete 'the' before stratospheric.
P2 L7. Remove comma after 'both'.
P2 L24. Remove comma after 'both'.
P2 L26. 'Flask' should be plural.
P2 L33. Start of sentence does not read correctly. Suggest replacing 'got' with 'has been'.
P4 L26. 'Details' should be singular.
P5 L4. There should be a space between 'Figure' and '2'.
P5 L6. This sentence doesn't quite make sense. I think it's supposed to read 'bellows pumps, in this set-up *the samples* are pressurized…'.
P5 L15. Purely out of curiosity – how many samples can be obtained from a single whole air sample at a flow rate of 100/150 ml/min? It might be good to include this figure at some point in the text.

P6 L16. The mention of the presence of an air-conditioning unit in the laboratory is fitting given the nature of the gases studied. However, it would be more useful to quote the refrigerant blend used, e.g. R-410A. Very high levels of these gases in the laboratory environment might affect the analysis, if there are small leaks in the system.

P7 Figure 3 caption. I'm confused by what the solid black line represents. From the caption 'The solid black line represents the value expected from direct measurements of the standard gas and the synthetic air.' Why is the black line a mix of the two gases? I would have thought it would be better if the black line was a direct measurement of the undiluted standard gas? Or is this to account for contamination in the synthetic air. Some additional description would be useful here.

P7 L9. Throughout the previous paragraph tetrachloroethene is referred to as '$C_2Cl_4$', but here it is referred to be its full name. Suggest using $C_2Cl_4$ throughout.

P8 L2. Suggest replacing 'right' with 'immediately'.

P8 L19. 'Is' should be 'are'.

P9 Figure 4. Some more explanation of the light blue trend in panel b) is required.

P9 Figure 4. There is no mention of HFC-152a (panel c) in the caption.

P9 L1. Why did HFC-152a increase over time?

P9 L3. Why did H-1301 increase over time?

P11 Figure 5. Are the ozone measurements from multiple instruments? What is the difference between the diamonds and the red line? It looks like one is an in-situ instrument and the other based on the flasks – either way, it would be good to include this information in the Figure caption.

P13 Figure 6. It would be useful to include the R value somewhere on each individual plot.

P13 Figure 6. There is some overlap of the colour bar with the underlying scatter plot.

---

## Referee Comment (RC2) · Anonymous Referee #2 · 20 Aug 2019

General notes: The tests of the stability of samples inside the HIRES sampler are welcome (though they should have been performed much earlier).

General: This paper adds valuable information about the stability of halocarbons measured sampled by the CARIBIC project using the HIRES sampler in the UT/LS. It reveals problems for many species with respect to general stability and/or reactivity towards ozone. These tests are welcome, though they should have been performed much earlier. Similar tests for other analyzed species, such as hydrocarbons would be of value as well.

In addition to the tests with artificial air mixtures, the HIRES sampler should be tested

under flight conditions. That is, several cylinders should be filled with the same UT/LS air and then analyzed over the course of days to weeks in the laboratory. Otherwise, the title should probably be changed to something like "Stability of Halocarbons in Simulated Air Samples from the Upper Troposphere and Lowermost Stratosphere". The authors have already some supporting evidence from actual flights; see CH2Cl2 in Figure 5, as well as Figures 6. This should be added to the discussion and table 1. Then the title could perhaps be retained.

Specific comments:

P. 3. Line 9: What kind of stainless steel is the sampler made out of? Is it electropolished? How was it welded?

P. 3. Line 13: How were the leak tests performed? Static with a gas? If so, using what gas at what pressure? Or is it evacuated? Why does the sampler contain either air from the previous flight or gas from the leak test? Also, have the authors considered to precondition the cylinders with moist air? This may have a positive impact on storage for several species (unless the water layer is removed quickly by the dry UT/LS air).

P. 3. Line 17: What final pressure is usually achieved after 20 s venting? In other words, what is the dilution factor? If tropospheric/laboratory air is still in the cylinders, more flushing is needed than if previous UT/LS samples are still in the cylinders.

P. 3. Line 18: Again, what is the final dilution? 0.2 to the 8? Is this dilution sufficient to flush out lower tropospheric/laboratory air?

P. 4. Line 1: The flushing/filling procedure should be explained in more details. Are the three flushing iterations in addition to the previous 8 times? Or are those the last three of the eight?

P. 4. Line 5: How much time does usually pass (min/max/mean) between sample taking in the airplane and analysis in the lab?

P. 4. Line 9: What is the effect of heating the Mg(ClO4)2 on the analytes?

Section 2.2: There does not seem to be any focusing step involved, other than on the pre-column, which at 50 oC probably does not focus very much. How sharp/wide are the peaks of the most volatile peaks? Have the authors considered to add a micro-focusing trap?

P. 4. Line 29: Please identify the individual Scripps scales for each compound somewhere. Please keep in mind scale revisions.

P. 4. Line 30: What are the calibration scales for CO and O3/how are their measurements calibrated?

Section 2.3: Does your Mg(ClO4)2 drying result in water vapor mixing ratios similar to those in the UT/LS? Please specify the dryness of the standard used for the experiments in comparison to UT/LS dryness. Keep in mind that the drier the samples, the more storage problems are likely to occur for certain halogenated compounds.

P. 5. Line 13: Were the HIRES cylinders flushed the same way as during flight? If not, how? What dilutions were achieved? Did you measure final water vapor in the HIRES cylinders?

P. 5. Line 15: Is one week the typical storage time for actual HIRES samples before halocarbon analysis? In the next sentence you indicate that storage time is usually much longer. Are your tests therefore representative of actual storage effects?

P. 6. Line 6: Have you considered to sample six cylinders at a time, thus increasing the volume and measuring an average storage effect rather than the storage effects in individual cylinders? If you had one rogue cylinder (which behaves much worse) could your tests identify it?

P. 6. Line 17: You cannot assume that synthetic air is free of halocarbons at the ppt level. I see that you have analyzed the zero air. Please rephrase the paragraph.

P. 6. Line 30ff: The text refers to C2Cl4, but figure 3b refers to CH2Cl2. Which one is it? Please check the correct chemical names/formulas throughout the text.

P. 7. Line 7: Why are no error bars for the individual data points shown? Please add them.

P. 7. Line 8ff and Figure 3: Why is the spread of the measurements on day 1 (and day 8) so much larger than the gray shaded area?

P. 8. Line 4ff: It seems straightforward that ozone reacts with any of the compounds containing double bonds, but I am very surprised that $CH_2Cl_2$ and $CH_3CCl_3$ (and $CHBr_2Cl$) were also depleted. $CH_2Cl_2$ for example is considered to be inert in organic chemistry. It is used as an inert solvent for ozonolysis of other compounds. Of course, we are talking about very different concentrations, but I just do not see how ozone reacts with $CH_2Cl_2$ or $CH_3CCl_3$. Do you have any explanation for this? Could you think of any other experimental problem for these compounds? $CH_2Cl_2$ for example, shows a very variable behavior in Figure 4. Have you repeated the storage tests to see if they are reproducible? The fact that $CH_2Cl_2$ is depleted in Figure 5 for all high-ozone periods is compelling, however. Do the other compounds which are affected by ozone also show this (consistently) during actual post-flight analysis? If so, please discuss and add another column to table 1.

P. 8. Line 12ff: It is known that $CH_3Cl$ and $CH_3Br$ may grow in stainless steel cylinders if they were not filled using particle filters. Whether this is due to sea salt or organic material or other compounds is unclear. Are the HIRES samples filled through a fine particulate filter?

P. 8. Line 4ff: Which of the observed effects in the ozone experiment do not agree with the long-term storage tests? In other words, could some of the "ozone" problems be general "storage" problems? If so, please discuss this.

Figure 3 and 4: Please add error bars. Figures for all compounds should be shown in the Supplement.

P. 8. Line 22: Please show error bars. Without error bars, the reader cannot put the

scatter of the results into perspective.

P. 8. Line 22ff: "some scatter" is unscientific. Also, without error bars, the reader cannot evaluate your statement about 2 sigma agreement.

P. 8. Line 24ff: Do you have an explanation why the red and blue experiments are so different? The red, yellow, blue, and dashed brown experiments indicate stability of CH2Cl2, while the other experiments show problems. Is it possible that individual cylinders are worse than others? How would you test that? Please expand your discussion at the end of page 8.

P. 8. Line 28: How do you determine two weeks (rather than one week)?

P. 9. Line 4: What polymer materials are used inside the sampling and the analytical system? Is Viton used by any chance? This could explain HFC-23 increases.

Figure 5 and 6: Please add a discussion of evidence for the observed storage tests from actual flights to the discussion. Please add Figures 5 and 6 for all compounds to the Supplement.

Figure 5 and 6: Please add correlation lines and R2 values (excluding the tropospheric outliers for CO) to Figures 5 and 6.

P. 12. Lines 6ff: I do not think that the discussion of correlations for an unstable compound is very informative.

P. 12. Line 30: I think this identifies a general weakness of CARIBIC. Can this be improved? Do the results from this paper have any implications for previously published results?

P. 15. Line 5ff: It is good to learn that a new sampler is being designed taking into account the lessons learned from this paper and that more rigorous tests will be performed.

Table 1: Can you exclude that the change of mixing ratios for drifting compounds is NOT

caused by drift of the working standard itself? In other words, do you have evidence for stability of the working standard for all compounds over the relevant time scales?

Minor comments:

P. 1. Line 21: Strike out "the". It should say "responsible for stratospheric ozone depletion".

P. 1. Line 23: ".. as an entry point for chlorinated and brominated species into the stratosphere".

P. 2. Lines 1/2: The trace gas composition in the ... can be analyzed ... or using air sample collection ...".

P. 2. Line 10: Even CO2 is not stable in all cylinders.

P. 2. Line 23: Flights take place over ...

P. 2. Lines 33ff: It has been regularly deployed since 2010 for post-flight measurements of greenhouse ...

P. 3. Line 1: Please add a few citations.

P. 3. Line 9: HIRES has been defined before.

P. 4. Line 16: What is 2 x 2 L reference volume? 4 L?

P. 4. Line 18: Please provide the supplier of the helium and the grade.

P. 4. Line 30: Strike out "in contrast".

P. 5. Line 5: This sentence is not quite right. Consider changing to "Contrary to ..., when the HIRES cylinders are filled with ambient air pressurized by ..., in this setup, the HIRES cylinders are ...".

P. 9. Line 3: "This also occurred for ...".

Table 1: The short-term column should be before the long-term column.

Figure 1: The CAD drawing of the sampler is nice, but I would also (perhaps rather) like to a drawing of the flow path.

[Figure]

---

## Author Comment (AC1) · 14 Oct 2019

**Stability of Halocarbons in Whole Air Samples**
**Response to Referee #1**

We thank referee#1 for the thorough reading of our manuscript, in particular for spotting the mismatch between the discussed compounds and compounds shown in Figure 3. All suggested changes and raised questions were carefully considered when revising the manuscript and we address them point by point in the following.

*Main comments:*

– *Figure 3. There seems to be some confusion regarding which species are presented. Panel B shows results for dichloromethane, whereas the text talks about tetrachloroethene (P6 L30, P7 L3). I'm assuming this is simply a mix-up, but it makes the entire section rather difficult to understand.*

Tetrachloroethene was mentioned in the text erroneously as the substance shown is dichloromethane. We have corrected this.

– *With regards to Figure 3, it is also not clear to me as to why the dichloromethane/CFC-12 ratio of the 'orange' sample is higher than the other two samples filled with the exact same ratio of standard/synthetic gas. I suspect that all three are within the measurement uncertainty. In which case it might be good to find some way of showing the error bars which are currently omitted.*

We have modified the figures for the revision of the manuscript now including daily values of the measurement precision for each individual day of measurement.

– *While there were several very interesting results presented, I did not feel that the authors provided much in the way of reasoning for a number of their findings. For instance, why did bromomethane and chloromethane increase in the cylinders over the length of the short-term (and long-term) storage tests? Presumably due to production as a result of the decomposition of more complex chlorinated compounds? The same can be said of H-1301, which would appear to be a strange result. I would not expect to see complex chemical mechanisms in this sort of paper, but it would be useful to have some brief discussion of these points.*
*We think that the current experiments do not provide sufficient information to speculate about the mechanisms of the observed changes and this is beyond the scope of the experiment series. We have, however, seen growth of chloromethane and chloromethane in newly welded cylinders of identical shape to the ones used inside HIRES. In addition, we have seen both compounds to be unstable in a high-pressure stainless steel gas bottle, though on longer time periods of the order of months to years. The result for H-1301 is indeed strange and we will look into this further with new cylinders in future experiments with and without the multiposition valves.*

*Technical corrections:*

– *I think the title needs to be revised. At current, it suggests that the storage tests were conducted on samples collected in the UTLS, while in fact they were conducted using samples prepared in the laboratory.*

We agree and have modified the title which now reads: "*Stability of Halocarbons in Whole Air Samples Collected in Stainless Steel Canisters* "

– *P1 L1. 'Halogenated halocarbons of' should be changed to 'halocarbons in'.*
Changed to: `Measurements of halogenated trace gases in ambient air …`

– *P1 L17. The sentence needs extending. Suggest 'growth was observed during storage for some compounds,…'*
*The sentence is modified and now reads:* "`Also growth was observed during storage for some species, namely for HFC-152a, HFC-23, and Halon-1301.`"

– *P1 L21. Delete 'the' before stratospheric.*
Done.

– *P2 L7. Remove comma after 'both'.*
– *P2 L24. Remove comma after 'both'.*
Changed.

– *P2 L26. 'Flask' should be plural.*
Done.

– *P2 L33. Start of sentence does not read correctly. Suggest replacing 'got' with 'has been'.*
Changed.

– *P4 L26. 'Details' should be singular.*
Changed.

– *P5 L4. There should be a space between 'Figure' and '2'.*
Modified.

– *P5 L6. This sentence doesn't quite make sense. I think it's supposed to read 'bellows pumps, in this set-up the samples are pressurized…'.*
The sentence was changed to: "`Contrary to the set-up in flight, when the HIRES cylinders are filled with ambient air pressurized by the metal bellows pumps, in this set-up they are pressurized directly from high pressure gas cylinders.`"

– *P5 L15. Purely out of curiosity – how many samples can be obtained from a single whole air sample at a flow rate of 100/150 ml/min? It might be good to include this figure at some point in the text.*

If the samples were exclusively for analysis in our lab, we could obtain 3 enrichments of 1 l from a 4.5bar cylinder (which is 3.5bar above ambient; at pressures below 0.5 bar above ambient the flow rate starts to decrease significantly and we found that measurements with very low enrichment flows are not reliable). From samples from our ground site program we usually do 2 enrichments for quality assurance. This cannot be done with HIRES samples because the air needs to be saved for other analyses. We prefer to not discuss this in the manuscript as it is unrelated to the question of compound stability.

– *P6 L16. The mention of the presence of an air-conditioning unit in the laboratory is fitting given the nature of the gases studied. However, it would be more useful to quote the refrigerant blend used, e.g. R-410A. Very high levels of these gases in the laboratory environment might affect the analysis, if there are small leaks in the system.*

The air conditioning of the laboratory uses water, thus it can be excluded that the results are influenced. We have added the following statement in the revised version of the manuscript: "`The air conditioning uses water and is therefore not expected to adversely influence halocarbon analysis.`"

– *P7 Figure 3 caption. I'm confused by what the solid black line represents. From the caption 'The solid black line represents the value expected from direct measurements of the standard gas and the synthetic air.' Why is the black line a mix of the two gases? I would have thought it would be better if the black line was a direct measurement of the undiluted standard gas? Or is this to account for contamination in the synthetic air. Some additional description would be useful here.*

*We look at ratios of all discussed compounds relative to CFC-12, and the black line does indeed represent that ratio for the undiluted standard gas. However, for those compounds which are present as contamination in the synthetic air, i. e. carbonyl sulfide, chloromethane, HFC-152a, and tetrachlorethene, this contamination has to be - and is – taken into account. We have removed that somewhat confusing statement here and refer to the explanation in section 2.3 instead.*

– *P7 L9. Throughout the previous paragraph tetrachloroethene is referred to as 'C2Cl4', but here it is referred to be its full name. Suggest using C2Cl4 throughout.*

*As noted above the figure shows dichloromethane while the text said tetrachlorethene (for which results are similar). This is corrected in the revised version of the manuscript.*

– *P8 L2. Suggest replacing 'right' with 'immediately'.*

Changed to: "`already one day after pressurization`"

– *P8 L19. 'Is' should be 'are'.*

Changed to "`shown is the ratio`" as in the figure caption.

- *P9 Figure 4. Some more explanation of the light blue trend in panel b) is required.*

  The corresponding paragraph now reads:
  "*Some substances that were found to be stable during the one week short-term test decreased after longer storage times, for example dichloromethane shown in Figure 4(b). A similar behaviour was observed for trichloromethane (CHCl3), tetrachloromethane (CCl4), trichloroethene (C2HCl3), tetrachloroethene (C2Cl4), tribromomethane (CHBr3), and bromochloromethane (CH2BrCl). Measurements of these compounds should not be evaluated for the HIRES canisters if analysis takes place later than two weeks after sample collection as the long-term test indicates changes of mixing ratios start to occur after that period. In general, the decrease during long-term seems to be independent of the influence of ozone, although the gas mixture that shows the largest depletion (light blue dashed line, mostly cut off in Figure 4) did contain ozone. Ozone could not be monitored during the pressurization of the samples. It can therefore not be excluded that this gas mixture may have been exposed to a different amount of ozone than the one represented by the dashed brown line which could have caused the stronger depletion.*"

- *P9 Figure 4. There is no mention of HFC-152a (panel c) in the caption.*
  This was added.

- *P9 L1. Why did HFC-152a increase over time?*
- *P9 L3. Why did H-1301 increase over time?*
  Currently we do not have any explanation for this. As we speculate in the original manuscript, a possible source of fluourinated compounds could be valve sealing material as the seals of the Valco multiposition valves are made from PFTE and currently we cannot add any further reasoning on why these compounds grow.

  We hope that further tests with new cylinders with and without this type of valves will shed some light on this growth. The result for Halon-1301 is indeed confusing. In an experiment performed with newly welded cylinders only using stainless steel parts (including the valves) we did not observed measurable changes of Halon-1301 over a storage period of 12 days. We will perform further experiments with and without the multiposition valves.

- *P11 Figure 5. Are the ozone measurements from multiple instruments? What is the difference between the diamonds and the red line? It looks like one is an in-situ instrument and the other based on the flasks – either way, it would be good to include this information in the Figure caption.*
  *The diamonds are the results of the fast in-situ measurement integrated of the sample period of the canister samples. This information is added to the figure caption which in the revised manuscript reads: "*Time series of ozone (red), CFC-12 (yellow) and dichloromethane (blue) during a flight from Munich to Denver on 22 March 2018. Ozone high resolution data represented by the red line were

```
integrated over the sampling period of each whole air
sample (red diamonds)."
```

− *P13 Figure 6. It would be useful to include the R value somewhere on each individual plot.*
R² values were added to the plots.

− *P13 Figure 6. There is some overlap of the colour bar with the underlying scatter plot.*
This is on purpose to not interfere with axis labels.

---

## Author Comment (AC2) · 14 Oct 2019

**Stability of Halocarbons in Whole Air Samples**
**Response to Referee #2**

We thank referee#2 for the thorough reading of our manuscript and we appreciated the helpful comments and remarks. During revision of the manuscript we considered all suggested modifications and questions which are addressed point by point in the following.

*General notes:*

–   *General: This paper adds valuable information about the stability of halocarbons measured sampled by the CARIBIC project using the HIRES sampler in the UT/LS. It reveals problems for many species with respect to general stability and/or reactivity towards ozone. These tests are welcome, though they should have been performed much earlier. Similar tests for other analyzed species, such as hydrocarbons would be of value as well. In addition to the tests with artificial air mixtures, the HIRES sampler should be tested under flight conditions. That is, several cylinders should be filled with the same UT/LS air and then analyzed over the course of days to weeks in the laboratory.*

We fully agree that a storage test with parallel filling of several cylinders in flight would yield valuable results. Currently, parallel filling of several samples is technically not possible without major reprogramming of the control unit software. The reason is that to avoid cross-contamination of sample in flight only one valve may be open at a given time. Therefore, while easily possible on the ground, such a test cannot be performed in flight.

–   *Otherwise the title should probably be changed to something like "Stability of Halocarbons in Simulated Air Samples from the Upper Troposphere and Lowermost Stratosphere".*

We agree that the initially chosen title does not accurately reflect the content of the manuscript. The title of the revised manuscript will read "*Stability of Halocarbons in Whole Air Samples Collected in Stainless Steel Canisters* ".

*Specific comments:*

–   *P. 3. Line 9: What kind of stainless steel is the sampler made out of? Is it electropolished? How was it welded?*
The canisters of the present HIRES were not electropolished. They are made from stainless steel (standard 1.4541), welding was micro plasma welding. Canisters for a new sampling unit currently under construction are made from electropolished steel and are electron beam welded in a vacuum chamber.

–   *P. 3. Line 13: How were the leak tests performed? Static with a gas? If so, using what gas at what pressure? Or is it evacuated? Why does the sampler contain either air from the previous flight or gas from the leak test? Also, have the authors considered to precondition the cylinders with moist air? This may have a positive impact on*

*storage for several species (unless the water layer is removed quickly by the dry UT/LS air).*

Pre-flight leak tests are performed with ambient air passed through a molecular sieve. Evacuation of the cylinders is not possible, therefore there is always air from the previous filling left. The reason is that the canisters are made from stainless steel foil of only 0.25mm thickness. They are mechanically stabilized but nevertheless will fully collapse when evacuated. Tests during the construction phase showed that evacuation below 600 mbar is not save. Preconditioning with moist air was not tested so far.

The corresponding section of the revised manuscript now reads: "`Before a flight, HIRES undergoes leak testing with ambient air passed through a molecular sieve, but cylinders are not preconditioned. On take-off, cylinders will usually hold remnant air from the last research flight or from the leak test. The reason is that due to mechanical stability of the thin-walled flasks they should not be evacuated to absolute pressures below 600 mbar.`"

‒ *P. 3. Line 17: What final pressure is usually achieved after 20 s venting? In other words, what is the dilution factor? If tropospheric/laboratory air is still in the cylinders, more flushing is needed than if previous UT/LS samples are still in the cylinders.*
‒ *P. 3. Line 18: Again, what is the final dilution? 0.2 to the 8? Is this dilution sufficient to flush out lower tropospheric/laboratory air?*

After 20s venting, ambient pressure is reached which aboard the aircraft is approx. 700 mbar, resulting in a dilution factor of less than 0.2. Tests during the construction phase and monitoring based on NMHC measurements during the first years of operation of the sampler have shown that eight iterations of flushing do reliably dilute remnants of previous fillings of tropospheric air.

The new wording in the revised manuscript reads: "`Tests during the construction phase and monitoring based on NMHC measurements during the first years of operation of the sampler have shown that eight iterations of flushing do reliably dilute remnants of previous fillings of tropospheric air. In flight, canisters are therefore flushed with ambient air eight times, this is achieved by filling a flask to 4 bar followed by venting for 20 s. After this time ambient pressure is reached which aboard the aircraft at flight altitude is 700 mbar. After that, canisters are eventually pressurized to 4.5 bar. The total time needed for this procedure is 4min of which the final pressurization  takes 10-20s.`

‒ *P. 4. Line 1: The flushing/filling procedure should be explained in more details. Are the three flushing iterations in addition to the previous 8 times? Or are those the last three of the eight?*

They are the last of the eight and this has been made more clear in the revised version of the manuscript. P4-L1-3 now read: "`The sampling period is defined as the time interval during which at least 97 % of`

*the sample air was collected. This comprises the last three of the eight flushing iterations and the final pressurization stage, adding up to a total sampling time of 1—2 min."*

– *P. 4. Line 5: How much time does usually pass (min/max/mean) between sample taking in the airplane and analysis in the lab?*
To answer this question, the following statement was added during revision of the manuscript at the end of subsection 2.1.: "*If a halocarbon analysis is performed it is usually last in a series of measurements and takes place approximately 3 to 5 weeks after the flight. The duration of the long-term storage test time of 8 weeks was deliberately chosen beyond this period."*

– *P. 4. Line 9: What is the effect of heating the Mg(ClO4)2 on the analytes?*

All the tubing is heated to avoid condensation of moisture (relevant for HIRES only for tropospheric samples). In addition, for example bromoform tends to get lost to walls of tubing if these are not heated. We have added the following statement to the text: "*All tubing is heated to avoid condensation of moisture (relevant for HIRES only for tropospheric samples) and to minimize wall losses."*

– *Section 2.2: There does not seem to be any focusing step involved, other than on the pre-column, which at 50 oC probably does not focus very much. How sharp/wide are the peaks of the most volatile peaks? Have the authors considered to add a microfocusing trap?*
At this stage we have not considered adding such a trap as satisfactory measurement precisions are reached. Peak width of the most volatile compounds is typically around 15 seconds (FWHM). At the higher retention times SIM windows are very clean und do usually contain only one peak

– *P. 4. Line 29: Please identify the individual Scripps scales for each compound somewhere.*
– *Please keep in mind scale revisions.*
Information on individual scales for each compound we think can be omitted here, as the results of the storage test do not depend on it as long as the reference gas and the measurements are calibrated on the same scale. In general, scale revisions are taken into account by close collaboration of our lab with the AGAGE network which includes intercalibration and exchange of standards.

– *P. 4. Line 30: What are the calibration scales for CO and O3/how are their measurements calibrated?*
CO is calibrated in-flight at 25 minute intervals with an onboard calibration standard, results are reported on the most recent WMO scale, currently this is WMO CO-X2014A.
The ozone instrument combines two techniques: two-channel UV photometry and dry chemiluminescence detection. The UV photometer is regularly cross-checked to a laboratory standard (a long-path UV photometer standard, UMEG, Germany) which

was referenced to the WMO standard reference photometer (SRP) #15 at EMPA (Switzerland). The chemiluminescence detector is calibrated vs. the UV photometer in post processing.

These details were published by Scharffe et al. (2012) (CO) and Zahn et al. (2012) (Ozone) and are therefore not repeated in the current manuscript but we refer to these specialized publications by the following statement: "*Details of the respective calibration of both instruments were published by Scharffe et al. (2012) and by Zahn at al. (2012).*"

– *Section 2.3: Does your Mg(ClO4)2 drying result in water vapor mixing ratios similar to those in the UT/LS? Please specify the dryness of the standard used for the experiments in comparison to UT/LS dryness. Keep in mind that the drier the samples, the more storage problems are likely to occur for certain halogenated compounds.*

We are aware of the fact that storage issues might arise from the dryness of the samples. As mentioned in the manuscript a dry standard was deliberately chosen for the storage experiments to be comparable to samples from the UT, although stratospheric samples may be much drier.

Unrelated to the HIRES storage tests we found halomethanes to be instable even in a moist stainless steel gas cylinder (large volume, high pressure), likely depending on pressure.

– *P. 5. Line 13: Were the HIRES cylinders flushed the same way as during flight? If not, how? What dilutions were achieved? Did you measure final water vapor in the HIRES cylinders?*

They were flushed the same way as during flight and this point was added in the revised version of the manuscript. Final water vapor was not measured. The standard used had a water vapor content of approx. 200ppm, thus taking into account dilution with dry synthetic air samples would have contained less than this value.

– *P. 5. Line 15: Is one week the typical storage time for actual HIRES samples before halocarbon analysis? In the next sentence you indicate that storage time is usually much longer. Are your tests therefore representative of actual storage effects?*

Typically, the halocarbon analysis takes place approx. 3 to 5 weeks after the flight. The long-term storage test time of 8 weeks was chosen to safely cover this period. The short-term storage test is thus representative for routine operation, but turned out to be crucial for the discussion of the possible influence of ozone.

– *P. 6. Line 6: Have you considered to sample six cylinders at a time, thus increasing the volume and measuring an average storage effect rather than the storage effects in individual cylinders? If you had one rogue cylinder (which behaves much worse) could your tests identify it?*

A laboratory storage test could only identify a cylinder behaving systematically different (for example because of the quality of its individual welding seams) when repeated several times.  Long-term an individual odd cylinder would be detected by evaluating measurement results based on canister number.  As regular halocarbon measurements have just started, current results do not give indication of any such

behavior, however, we will pay close attention to this issue with the number of measurements increasing.

- *P. 6. Line 17: You cannot assume that synthetic air is free of halocarbons at the ppt level. I see that you have analyzed the zero air. Please rephrase the paragraph. P. 6. Line 30ff: The text refers to C2Cl4, but figure 3b refers to CH2Cl2. Which one is it? Please check the correct chemical names/formulas throughout the text.*
- *P. 7. Line 7: Why are no error bars for the individual data points shown? Please add them.*

We refrain from adding error bars to the individual data points because they overlap and merge into one big error bar for each day of measurement. To compensate for this lack of information the 2-sigma band around the expected value was included. As this came out not sufficiently clear, we have modified the figures.  In contrast to the previous version, figures now include the precision of each measurement day rather than the average precision used before. This accounts for the varying measurement uncertainty of our GC-MS system.

- *P. 7. Line 8ff and Figure 3: Why is the spread of the measurements on day 1 (and day 8) so much larger than the gray shaded area?*
The measurement precision achieved has daily variations for example depending on the strength of the instrumental drift during a measurement day or ageing of the sample loop. This is reflected in the observed scatter but was not taken into account by the grey shaded area which was based on an average instrumental precision value. This has now been included in the revised version of the figures by showing the daily precision of each measurement days rather than an average precision as before. If the scatter is larger that the instrumental drift on a measurement day (as derived from the measurements of the laboratory standard) it points to variability of the sample mixing ratio.

- *P. 8. Line 4ff: It seems straightforward that ozone reacts with any of the compounds containing double bonds, but I am very surprised that CH2Cl2 and CH3CCl3 (and CHBr2Cl) were also depleted. CH2Cl2 for example is considered to be inert in organic chemistry. It is used as an inert solvent for ozonolysis of other compounds. Of course, we are talking about very different concentrations, but I just do not see how ozone reacts with CH2Cl2 or CH3CCl3. Do you have any explanation for this? Could you think of any other experimental problem for these compounds? CH2Cl2 for example, shows a very variable behavior in Figure 4.*
Please not that CH3CCl3 was listed erroneously in this line which will be corrected when revising the manuscript.
We do not have an explanation for the depletion of CH2Cl2 and CHBr2Cl. An additional experiment performed in the meantime indicated that both substances could not reliable be retrieved after filling cylinders constructed identically to the ones used in HIRES directly from the reference gas in a much simpler setup with no ozone involved. We can currently not speculate by which mechanism the observed depletion could have been caused. We think that Figure 3b shows convincing evidence that ozone did have an impact, but there likely are additional issues.

- *Have you repeated the storage tests to see if they are reproducible?*

The time-consuming tests could only be performed during a longer operational break of CARIBIC flights in 2016. They could not be repeated since, because this would cause an unacceptable long grounding of the CARIBIC container. For legal reasons, the air sampler has to be part of the instrument package during each flight of the container, even if it was not operated. It can therefore not be removed for laboratory test for the necessary time.

– *The fact that CH2Cl2 is depleted in Figure 5 for all high-ozone periods is compelling, however. Do the other compounds which are affected by ozone also show this (consistently) during actual post-flight analysis? If so, please discuss and add another column to table 1.*

Other compounds affected by ozone in the storage tests do consistently show a behavior similar to dichloromethane.  A corresponding statement is included in the revised version of the manuscript and the paragraph now closes: "`A similar behavior was found for tetrachloroethene and trichloromethane and for the tropospheric samples for trichloroethene and for dibromochloromethane. The latter compound was below its detection limit in all stratospheric samples, trichlorethene in several of them.`"

Please note that previously tetrachloromethane was erroneously marked as influenced by ozone in Table 1.

– *P. 8. Line 12ff: It is known that CH3Cl and CH3Br may grow in stainless steel cylinders if they were not filled using particle filters. Whether this is due to sea salt or organic material or other compounds is unclear. Are the HIRES samples filled through a fine particulate filter?*

They are filled through a 2µ filter (Swagelok SS-4F-2). This information has been added in the revised version of the manuscript into subsection 2.1.

– *P. 8. Line 4ff: Which of the observed effects in the ozone experiment do not agree with the long-term storage tests? In other words, could some of the "ozone" problems be general "storage" problems? If so, please discuss this.*

Ozone problems were diagnosed from the measurement that took place right after filling of the samples, therefore they are unlikely to result from storage.

When thinking about the consequences for real UT-LS data, it has to be taken into account that in our experiment the reference gas is mixed with the ozone enriched synthetic air during the filling procedure, right before pressurization of the sample. In flight, stratospheric air masses with high ozone levels will be at some state of mixing and in a continuous chemically processing. A corresponding statement is included in the revised version of the manuscript: "`It should, however, be noted that the experiment does not adequately mimic stratospheric conditions. In the laboratory tests presented here, the reference gas is mixed with the ozone enriched synthetic air during the filling procedure. In flight, stratospheric air masses with high ozone levels will be at some state of mixing and in a continuous`

*chemically processing. In addition contact with hot*
*surfaces such as inside the metal bellows pumps will*
*destroy ozone."*

- *Figure 3 and 4: Please add error bars. Figures for all compounds should be shown in the Supplement.*
- *P. 8. Line 22: Please show error bars. Without error bars, the reader cannot put the scatter of the results into perspective.*

Addition of error bars does not add readable information to the figures as they overlap and merge into one big error bar for each day of measurement. However, we have modified the figures such that the 2-sigma range around the expected value - which reflects the combination of error bars on the expected value an the data point – to make it better visible. All points falling into the indicated band agree with the expected value within the measurement precision.

The corresponding part of the text now reads: "*For each measurement day a daily precision value was calculated from the variability of the measurements of the standard. This expected daily uncertainty range is represented by the grey error bars.*
*Solid coloured lines stand for canisters pressurized with synthetic air not treated by the UV-lamp, dashed coloured lines represent canisters pressurized with the synthetic air passing the UV-lamp and thus ozone being present. Error bars for the individual data points are not shown as they overlap and merge into one undistinguishable error bar. However, if the symbols fall within the uncertainty range indicated by the grey error bars of the expected value this means, that they agree within 2$\sigma$ with the expected value."*

We refrain from adding a supplement to this paper as we think Table 1 sufficiently summarizes the results.

- *P. 8. Line 22ff: "some scatter" is unscientific. Also, without error bars, the reader cannot evaluate your statement about 2 sigma agreement.*

The statement refers to the 2-sigma band indicated by he grey shaded area in the figures. As this fact did not come out very well, we have made it more clear in the text. The phrase was reworded to: "*HFC-134a variability is smaller than the measurement precision and measured mixing ratios agree within 2$\sigma$ with the expected value."*

- *P. 8. Line 24ff: Do you have an explanation why the red and blue experiments are so different? The red, yellow, blue, and dashed brown experiments indicate stability of CH2Cl2, while the other experiments show problems. Is it possible that individual cylinders are worse than others? How would you test that? Please expand your discussion at the end of page 8.*

We do not have an explanation why the measurements indicated by the blue and solid red lines deviate so much from each other, which they do in the case of dichloromethane the most on the last two measurement days.  However, compared

to the uncertainty range of the individual days of measurement (cf. revised figures) CH2Cl2 seems not stable also in the other canisters.

It seems likely that individual canisters may perform worse or better than others and we will pay close attention to the behavior of individual canisters when analyzing flight samples, as this should become visible over the course of regular measurements.

We have added the following statement in the revised version of the manuscript: "*In general, the decrease during long-term storage seems to be independent of the influence of ozone, although the gas mixture that shows the largest depletion (light blue dashed line, mostly cut off in Figure 4) did contain ozone. Ozone could not be monitored during the pressurization of the samples. It can therefore not be excluded that this gas mixture may have been exposed to a different amount of ozone than the one represented by the dashed brown line which could have caused the stronger depletion.*"

- *P. 8. Line 28: How do you determine two weeks (rather than one week)?*

The two weeks are estimated from the results of the long-term storage test which indicate that changes to the initial mixing ratios start to become relevant after this time.

The sentence is rephrased to: "*Measurements of these compounds should not be evaluated for the HIRES canisters if analysis takes place later than two weeks after sample collection as the long-term test indicates changes of mixing ratios start to occur after that period.*"

- *P. 9. Line 4: What polymer materials are used inside the sampling and the analytical system? Is Viton used by any chance? This could explain HFC-23 increases.*

During sample enrichment all connectors and tubing used are made from stainless steel with exception of two Valco valves in the sample flow. These do contain PFTE seals (similar to the ones used in HIRES) but this cannot explain increases occurring in the test samples, because during storage those were not connected to the analytical system. During the measurement, contact times with the valves are minimized and all lines are flushed prior to enrichment.

- *Figure 5 and 6: Please add a discussion of evidence for the observed storage tests from actual flights to the discussion.*

The discussion in section 3.3 has been extended in the revised version of the manuscript. In particular, the following statements were added or reworded to:

- *CFC-12 anticorrelates with mixing ratios of ozone and this is also found for the other long-lived compounds which were stable in the storage experiments. Such a behaviour is expected, because ozone-rich stratospheric air masses are aged and should contain lower mixing ratios depending on a substance's stratospheric lifetime and transport pathway. Three of the canisters analysed from this flight were collected in tropospheric air masses characterized by*

*lower mixing ratios of ozone levels. Mixing ratios of CFC-12 measured in these samples are around 510 ppt, consistent with current tropospheric mixing ratios observed at ground sites (Schuck et al., 2018). Similarly consistent numbers are measured for the other compounds expected to be stable in the canisters according to the storage experiments.*

- *In the tropospheric air samples, dichloromethane varied between 14ppt and 49ppt. This agrees with mixing ratios in tropospheric samples in the dataset presented by Leedham-Elvidge et al. (2015) which were up to 65ppt with an increase observed from 2006 through 2012, but is somewhat lower than mixing ratios at the ground in March 2018 (Schuck et al. 2018) which would be consistent with the result from the storage test, that dichloromethane is not stable in HIRES cylinders long-term.*

- *A similar behaviour was found for tetrachloroethene and trichloromethane and for the tropospheric samples for trichloroethene and for dibromochloromethane. The latter compound was below its detection limit in all stratospheric samples, trichloroethene in several of them.*

— *Please add Figures 5 and 6 for all compounds to the Supplement.*
It cannot be avoided that our selection of compounds is to some extent random and remains uncomplete. We don't agree that adding a supplement to the manuscript does add valuable information which is not contained in the summary Table 1 and therefore we prefer to not add supplementary information to the paper.

— *Figure 5 and 6: Please add correlation lines and R2 values (excluding the tropospheric outliers for CO) to Figures 5 and 6.*
As Figure 5 shows the exemplary time series of flight data, we assume this refers to the scatter plots in Figures 6 and 7. Following your suggestion, we prepared modified versions of Figures 6 and 7 including correlation lines and r² values. The vertical axis of Figure 7(c) was set to the same range as panels (a) and (b).

— *P. 12. Lines 6ff: I do not think that the discussion of correlations for an unstable compound is very informative.*
Although we cannot deduce direct conclusions from this correlation (such as emission ratios or time of chemical processing), we think it an interesting detail that there is a correlation at all. This is despite the compound being unstable during storage and the mixing ratios being depleted in the stratospheric samples.

— *P. 12. Line 30: I think this identifies a general weakness of CARIBIC. Can this be improved? Do the results from this paper have any implications for previously published results?*

We do not see that this could be improved, because no single lab among the collaborators has the capability nor the capacity to perform all different measurements (greenhouse gases, non-methane hydrocarbons, halocarbons, and possibly also isotopic composition analysis) in one place and within shorter times. Halocarbon measurements of HIRES samples were not published up to now, therefore there are no reverse implications on previous results from our tests.

– *P. 15. Line 5ff: It is good to learn that a new sampler is being designed taking into account the lessons learned from this paper and that more rigorous tests will be performed.*
As it turned out in the meantime that the geometry of the cylinders with only one line of tubing of a small diameter does not allow adding a coating on the inside after welding, we removed the statement on tests of a coating.

– *Table 1: Can you exclude that the change of mixing ratios for drifting compounds is NOT caused by drift of the working standard itself? In other words, do you have evidence for stability of the working standard for all compounds over the relevant time scales?*
We can exclude drift of the working standard, because it is regularly compared to a tertiary AGAGE standard during regular sample measurements performed with the GC-MS setup.  A corresponding statement is included in the measurement section revised version of the manuscript: "`HIRES samples are measured relative to a laboratory standard which has been collected cryogenically at Jungfraujoch (Switzerland) in December 2007. It is compared to a tertiary standard of the Advanced Global Atmospheric Gases Experiment (AGAGE) network monthly and has been re-calibrated versus several AGAGE standards in December 2018. Drift of the working standard can thus be excluded.`"

*Minor comments:*

– *P. 1. Line 21: Strike out "the". It should say "responsible for stratospheric ozone depletion".*
Changed.

– *P. 1. Line 23: ".. as an entry point for chlorinated and brominated species into the stratosphere".*
Changed.

– *P. 2. Lines 1/2: The trace gas composition in the ... can be analyzed ... or using air sample collection ...".*
Changed.

– *P. 2. Line 10: Even CO2 is not stable in all cylinders.*
Since a discussion of CO2 is beyond the scope of this manuscript that sentence will be removed in the revised version.

- *P. 2. Line 23: Flights take place over ...*
  Changed to "Measurement flights".

- *P. 2. Lines 33ff: It has been regularly deployed since 2010 for post-flight measurements of greenhouse ...*
  Changed.

- *P. 3. Line 1: Please add a few citations.*
  References Navarro et al. 2015 and Keber at al. 2019 were added.

- *P. 3. Line 9: HIRES has been defined before.*
  The repetitive definition has been removed.

- *P. 4. Line 16: What is 2 x 2 L reference volume? 4 L?*
  The reference volume consist of two canisters wit a volume of 2 L each. Therefore, we prefer to write "2 x 2 L" over "4 L".

- *P. 4. Line 18: Please provide the supplier of the helium and the grade.*
  Helium grade 6.0 supplied by Praxair is used, this information has been added to the text.

- *P. 4. Line 30: Strike out "in contrast".*
  Done.

- *P. 5. Line 5: This sentence is not quite right. Consider changing to "Contrary to ..., when the HIRES cylinders are filled with ambient air pressurized by ..., in this setup, the HIRES cylinders are ...".*
  Changed.

- *P. 9. Line 3: "This also occurred for ...".*
  Changed.

- *Table 1: The short-term column should be before the long-term column*
  We agree and the table is changed correspondingly in the revised manuscript.

- *Figure 1: The CAD drawing of the sampler is nice, but I would also (perhaps rather) like to a drawing of the flow path.*
  We have rephrased the figure caption to make clear, that the CAD drawing is deliberately reduced to the main components, not showing any tubing and therefore not indicating sample air flow paths. We finally opted not to include a drawing of the flow path because we don't think it would add any information not contained in Figure 2.

**References:**

Keber, T., Bönisch, H., Hartick, C., Hauck, M., Lefrancois, F., Obersteiner, F., Ringsdorf, A., Schohl, N., Schuck, T., Hossaini, R., Graf, P., Jöckel, P., and Engel, A.: Bromine from short–lived source gases in the Northern Hemisphere UTLS, Atmos. Chem. Phys. Discuss., https://doi.org/10.5194/acp-2019-796, in review, 2019.

Leedham-Elvidge, E. C., Oram, D. E., Laube, J. C., Baker, A. K., Montzka, S. A., Humphrey, S., O'Sullivan, D. A., and Brenninkmeijer, C. A. M.: Increasing concentrations of dichloromethane, CH2Cl2, inferred from CARIBIC air samples collected 1998–2012, Atmospheric Chemistry and Physics, 15, 1939–1958, https://doi.org/10.5194/acp-15-1939-2015, 2015.

Maria A. Navarro, Elliot L. Atlas, Alfonso Saiz-Lopez, Xavier Rodriguez-Lloveras, Douglas E. Kinnison, Jean-Francois Lamarque, Simone Tilmes, Michal Filus, Neil R. P. Harris, Elena Meneguz, Matthew J. Ashfold, Alistair J. Manning, Carlos A. Cuevas, Sue M. Schauffler, Valeria Donets: Brominated compounds at the tropical tropopause, Proceedings of the National Academy of Sciences Nov 2015, 112 (45) 13789-13793; DOI: 10.1073/pnas.1511463112

Scharffe, D., Slemr, F., Brenninkmeijer, C. A. M., and Zahn, A.: Carbon monoxide measurements onboard the CARIBIC passenger aircraft using UV resonance fluorescence, Atmos. Meas. Tech., 5, 1753–1760, https://doi.org/10.5194/amt-5-1753-2012, 2012.

Schuck, T. J., Lefrancois, F., Gallmann, F., Wang, D., Jesswein, M., Hoker, J., Bönisch, H., and Engel, A.: Establishing long-term measurements of halocarbons at Taunus Observatory, Atmospheric Chemistry and Physics, 18, 16 553–16 569, https://doi.org/10.5194/acp-18-16553-2018, 2018.

Zahn, A., Weppner, J., Widmann, H., Schlote-Holubek, K., Burger, B., Kühner, T., and Franke, H.: A fast and precise chemiluminescence ozone detector for eddy flux and airborne application, Atmos. Meas. Tech., 5, 363–375, https://doi.org/10.5194/amt-5-363-2012, 2012.

---

## Author Response (AR2)

**Response to Editor's Comments**

We thank the editor for his comments on the revised manuscript. All points are addressed individually below.

*1) "Whole" should be removed from the title as this is not strictly true. I would suggest "Stability of halocarbons in simulated air stored in stainless steel canisters"*

The title was changed and now reads "Stability of halocarbons in air samples stored in stainless steel canisters". We prefer to not use the term "simulated" as the experiments were done using an ambient air standard which was diluted but otherwise unchanged, and we did not use an artificially prepared gas mixture which might be concluded from the word "simulated".

*2) P2 L 32 - spell out which GHGs it has been used for to save confusion with halocarbons as GHGs*

We have listed $CO_2$, $CH_4$, $N_2O$ and $SF_6$ here.

*3) P3 L15 - more details on the molecular sieve needed and conditions under which this is done. What is removed in this step?*

The molecular sieve was put in as a general precaution measure to prevent possible contamination of tubing with hydrocarbons. A 10X molecular sieve is used. During revision of the manuscript the phrase was modified and now reads:
`Before a flight, HIRES undergoes leak testing with ambient air passed through a 10X molecular sieve (8 Å pore diameter) to prevent contamination of tubing with hydrocarbons. Cylinders are not preconditioned.`

*4) Can you relate Figure 1 to Figure 2? The valves in Fig 1 are the valves labelled g in Figure 2. From Figure 2 there doesn't appear to be any references to the lettered labelling of the valves. If this is the case please remove unnecessary letter labelling from the figure.*

The labels were removed.

*5) Page 6 line 19 - use "L", not "l" - check throughout*

This was corrected.

*6) In relation to possible blank issues (initially raised in relation to the AC system) - were lab blanks measured as routine throughout this test period?*

Unfortunately, there were no routine measurements of lab air. The way the GC-MS system is set up would require major modifications to do this, namely installation of an additional inlet line and a pump. Currently with this GC-MS system we can only measure from sample volumes which are at a significant overpressure relative to ambient. If absolute sample pressure is below 1.5 bar, sample flows become to low, in particular for small volume

samples as the 1 L HIRES canisters.  To do lab air measurements requires to pressurize sample canisters inside the laboratory, however, this is not done regularly and it was not done during the HIRES test period.

*7) P7 L21 and elsewhere - there is description of the plotting detail within the actual text. This should be clear from the figure caption and the legend. Particularly on page 9 line 27 where colours are mentioned - some readers won't be able to connect a colour to a description e.g. I think this looks more orange than brown!*

Legends of Fig. 3 and 4 were modified (see answer below on point 8). To avoid specific references to individual colours other than black which might be interpreted differently by readers, the passage on P.9 was reworded to:

`In general, the decrease during long-term storage seems to be independent of the influence of ozone, although the gas mixture that shows the largest depletion did contain ozone. Ozone could not be monitored during the pressurization of the samples. It can therefore not be excluded that the two gas mixtures represented by dashed lines in  Figure 4 may have been exposed to  different amounts of ozone.`

*8) I think it would be good to reiterate in the figure caption of Fig 4 the difference in how sampling was done between the short term and long term experiments (different canisters or same canister). If the reader hasn't read the text they can then instantly make sense of the results.*

We have extended the caption of Fig. 4 (and analogously of Fig. 3) by adding the following:
Fig. 3: "`Colours represent different canisters filled with the gas mixture indicated in the legend, each canister was measured twice, i. e. 1 day after pressurization and again after 8 days of storage.`"
Fig. 4: "`Colours represent groups of canisters which were simultaneously filled with the gas mixture indicated in the legend. On each measurement day one canister of each group was analyzed.`"

*9) Why were % changes removed in Table 1? Given lack of supporting data these numbers might be quite informative? Table title needs changing if numbers removed.*

The % changes are put back.

*10) The last paragraph could do with more explaining e.g. welding under vacuum. Is this different to before and why is this an improvement? Will there be any opportunities to do a new set of tests with the new canisters any time soon?*
*11) Given point 10 can more detail be provided on the origin of the canisters used here? This detail is given in response to the reviewer 2 but not in the revised text*

The new canisters will be tested at different stages while putting together the new sampler and first tests have already been done.

The last paragraph was extended and now reads:

*Currently, we are in the process of constructing a second high resolution air sampler for use inside the CARIBIC container. Based on the measurements presented here, close attention will be given to the manufacturing of the stainless steel cylinders which will be made of electro-polished stainless steel foil and welding will be done under vacuum. In contrast to the canisters tested here which were micro plasma welded from similar stainless steel foil which was not electro-polished we expect to obtain cleaner inside surfaces and cleaner welding seams. During the construction of the new sampling unit more stability tests like those presented here will be performed.*

*12) In reply to reviewer 2 you explain why these tests are difficult to repeat. Some mention of this in the text would be worthwhile. It sounds like you really made the most of a period when the sampler was on a break from flights.*

To explain this we have added the following statement to the text at the end of the introduction:
*Because the test series included storage of the sampling unit over several weeks and different tests were performed, it is not possible to do such experiments in between the regularly scheduled research flights of the CARIBIC container. This would cause an unacceptable long grounding of all the instrumentation because for reasons of aircraft certification the air sampler has to be part of the instrument package during each flight. Therefore, we took opportunity of a longer operational break of CARIBIC flights in 2016.*

[revised manuscript text omitted]

expected. No systematic change from day 1 to day 8 was measured for either of the two compounds. While dichloromethane was stable over a storage time of one week, it was influenced by ozone and exhibited depleted mixing ratios in the canisters already one day after pressurization. It can thus not be reliably measured from HIRES canisters from the lowermost stratosphere.

Substances that were found to be depleted in HIRES canisters when pressurized in the presence of ozone were: dichloromethane ($CH_2Cl_2$), trichloroethene ($C_2HCl_3$), tetrachloroethene ($C_2Cl_4$), and dibromochloromethane ($CHBr_2Cl$). Carbonyl sulfide (COS) showed higher mixing ratios in samples that had been exposed to ozone. While there is clear evidence, that the mixing ratios of these substances are modified in the canisters when they are pressurized at elevated levels of ozone, it is not possible from these experiments to deduce an ozone threshold above which results become unreliable. We will thus consider all UTLS samples characterized as stratospheric by ozone mixing ratios above the respective ozone chemical tropopause, potential vorticity or low mixing ratios of nitrous oxide as not suited for post-flight analysis of these compounds in samples from the current HIRES sampling unit.

It should, however, be noted that the experiment does not adequately mimic stratospheric conditions. In the laboratory tests presented here, the reference gas is mixed with the ozone enriched synthetic air during the filling procedure. In flight, stratospheric air masses with high ozone levels will be at some state of mixing and in a continuous chemically processing. In addition contact with hot surfaces such as inside the metal bellows pumps will destroy ozone.

Among the substances influenced by ozone, dibromochloromethane ($CHBr_2Cl$) additionally showed decreasing mixing ratios already after one week of storage (depleted by 94 %), while bromomethane ($CH_3Br$, +46 %) and chloromethane($CH_3Cl$, +14 %) were found to grow. Trichloroethene ($C_2HCl_3$) exhibited a variability which did not allow to draw stringent conclusions in the short-term storage test. Tribromomethane ($CHBr_3$), which was unaffected by ozone, was depleted by 70 % after one week of storage. Table 1 summarizes these results.

**3.2 Long-term stability**

The long-term storage test comprised measurements of pressurized canisters after storage times of 1, 8, 15, 29, 51, and 57 days. While for the short-term test, individual canisters were measured on day 1 and day 8, this was not possible for the long-term test. For the long-term test, six cylinders were simultaneously filled, and on each measurement day, the next canister of such a series was measured. Thus, it cannot be fully excluded that stability might not only depend on the substance investigated, but it might be a feature of an individual canister, for example related to the quality of welding seams.

Figure 4 shows as an example results of the long-term storage test for HFC-134a, dichloromethane and HFC-152a. As before, shown is the ratio of mixing ratios of the respective substance and CFC-12. The black line indicates the expected ratio and the grey shaded area represents the experimental 2-$\sigma$ uncertainty range. Solid lines are for measurements of the gas mixture without ozone, dashed line for those with ozone. Error bars for individual data points are not shown. Like most long-lived halogenated tracers, HFC-134a variability is smaller than the measurement precision and measured mixing ratios agree within $2\,\sigma$ with the expected value.

Some substances that were found to be stable during the one week short-term test decreased after longer storage times, for example dichloromethane shown in Figure 4(b). A similar behaviour was observed for trichloromethane ($CHCl_3$), tetra-

[Figure]

**Figure 4.** Results of the ozone and long-term storage test for HFC-134a (a), dichloromethane (b), and HFC-152a (c). Shown is the ratio of mixing ratios of the respective compound relative to that of CFC-12 to cancel dilution uncertainties. For dichloromethane the vertical scale has been adjusted cutting off samples that were strongly depleted. The solid black line represents the value expected from direct measurements of the standard gas and the synthetic air. Shading indicates the 2-$\sigma$ experimental uncertainty range around the expected value, error bars of individual data points are omitted for clarity. Colours represent groups of canisters which were simultaneously filled with the gas mixture indicated in the legend. On each measurement day one canister of each group was analyzed.

chloromethane (CCl$_4$), trichloroethene (C$_2$HCl$_3$), tetrachloroethene (C$_2$Cl$_4$), tribromomethane (CHBr$_3$), and bromochloromethane (CH$_2$BrCl). 
[revised manuscript text omitted]